# Robust Conformal Prediction with a Single Binary Certificate

**Soroush H. Zargarbashi**
CISPA Helmholtz Center for Information Security
zargarbashi@cs.uni-koeln.de

**Aleksandar Bojchevski**
University of Cologne
bojchevski@cs.uni-koeln.de

## Abstract

Conformal prediction (CP) converts any model's output to prediction sets with a guarantee to cover the true label with (adjustable) high probability. Robust CP extends this guarantee to worst-case (adversarial) inputs. Existing baselines achieve robustness by bounding randomly smoothed conformity scores. In practice, they need expensive Monte-Carlo (MC) sampling (e.g. $\sim 10^4$ samples per point) to maintain an acceptable set size. We propose a robust conformal prediction that produces smaller sets even with significantly lower MC samples (e.g. 150 for CIFAR10). Our approach binarizes samples with an adjustable (or automatically adjusted) threshold selected to preserve the coverage guarantee. Remarkably, we prove that robustness can be achieved by computing *only one* binary certificate, unlike previous methods that certify each calibration (or test) point. Thus, our method is faster and returns smaller robust sets. We also eliminate a previous limitation that requires a bounded score function.

## 1 Introduction

Despite their extensive applications, modern neural networks lack reliability as their output probability estimates are uncalibrated (Guo et al., 2017). Many uncertainty quantification methods are computationally expensive, lack compatibility with black-box models, and offer no formal guarantees. Alternatively, conformal prediction (CP) is a statistical post-processing approach that returns prediction *sets* with a guarantee to cover the true label with high adjustable probability. CP only requires a held-out calibration set and offers a distribution-free model-agnostic coverage guarantee (Vovk et al., 2005; Angelopoulos & Bates, 2021). The model is used as a black box to compute conformity scores which capture the agreement between inputs $x$ and labels $y$. These prediction sets are shown to improve human decision-making both in terms of response time and accuracy (Cresswell et al., 2024). CP assumes exchangeability between the calibration and the test set (a relaxation of the i.i.d. assumption), making it broadly applicable to images, language models, etc. CP also applies on graph node classification (Zargarbashi et al., 2023; Huang et al., 2023) where uncertainty quantification methods are limited. However, exchangeability, and therefore the conformal guarantee, easily breaks when the test data is noisy or subjected to adversarial perturbations.

Robust conformal prediction extends this guarantee to worst-case inputs $\tilde{x}$ within a maximum radius around the clean point $x$, e.g. $\forall \tilde{x}$ s.t. $\|\tilde{x} - x\|_2 \leq r$. In the evasion setting, we assume that the calibration set is clean, and test datapoints can be perturbed. Building on the rich literature of robustness certificates (Kumar et al., 2020), recent robust CP baselines (Gendler et al., 2021; Zargarbashi et al., 2024; Jeary et al., 2024) use a conservative score at test time that is a *certified* bound on the conformity score of the clean unseen input. This maintains the guarantee even for the perturbed input since "if CP covers $x$, then robust CP certifiably covers $\tilde{x}$". However, the average set size increases, especially if the bounds are loose. The certified bounds can be derived through model-dependent verifiers (Jeary et al., 2024) or smoothing-based black-box certificates (Zargarbashi et al., 2024).

For the robustness of black-box models, an established approach is to certify the confidence score through randomized smoothing (Kumar et al., 2020), obtaining bounds on the expected smooth score. The tightness of these bounds depends on the information about the smooth score around the given input, e.g. the mean Yan et al. (2024), or the CDF Zargarbashi et al. (2024). Such methods: (i) assume the conformity score function has a bounded range, (ii) compute several certificates for

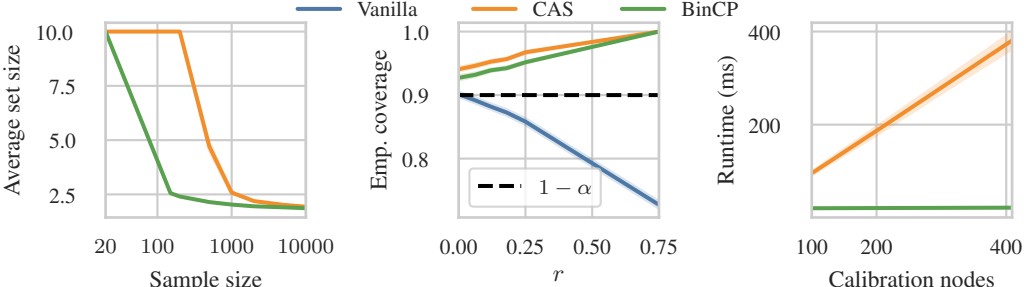

Figure 1: [Left] Average set size with different MC sample rates, [Middle] empirical coverage of vanilla and robust CPs under attack, and [Right] runtime of robust CP as a function of calibration datapoints (after computing the MC samples which is the number of lower bound computations).

each calibration (or test) point, and (iii) need a large number of Monte-Carlo samples to get tight confidence intervals. For the current SOTA method CAS (Zargarbashi et al., 2024), the accounting for sample correction inflates the prediction sets significantly for sample rates below 2000 (see Fig. 1-left). This inefficiency increases to trivially returning $\mathcal{Y}$ as the prediction set when we run with higher coverage rates or higher radii (see § 6). In contrast, we obtain robust and small prediction sets with only $\sim 150$ MC samples. Additionally, these methods require computing certified bounds for (at least) each calibration point which we further show is a wasteful computation.

**BinCP.** We observe that smooth inference is inherently more robust. Even without certificates, randomized methods show a slower decrease in coverage under attack (see Fig. 6-right). Given any score function $s(\boldsymbol{x}, y)$ capturing conformity, Zargarbashi et al. (2024) and Gendler et al. (2021) define the smooth score as $\bar{s}(\boldsymbol{x}, y) = \mathbb{E}_{\boldsymbol{\epsilon} \sim \mathcal{N}(\boldsymbol{0}, \sigma \boldsymbol{I})}[s(\boldsymbol{x} + \boldsymbol{\epsilon}, y)]$. Instead, we perform binarization via a threshold $\tau$, i.e. $\bar{s}(\boldsymbol{x}, y) = \mathbb{E}_{\boldsymbol{\epsilon} \sim \mathcal{N}(\boldsymbol{0}, \sigma \boldsymbol{I})}[\mathbb{I}[s(\boldsymbol{x} + \boldsymbol{\epsilon}, y) \geq \tau]] = \Pr_{\boldsymbol{\epsilon} \sim \mathcal{N}(\boldsymbol{0}, \sigma \boldsymbol{I})}[s(\boldsymbol{x} + \boldsymbol{\epsilon}, y) \geq \tau]$. Both are valid conformity scores, and both change slowly around any $\boldsymbol{x}$, however, our binarized CP (BinCP) method has several advantages. First, we define robust CP that only computes a single certificate. In comparison, CAS requires at least one certificate per calibration (or test) point. Second, our method can effortlessly use many existing binary certificates out of the box without any additional assumptions or modifications. A direct consequence is that we can use de-randomization techniques (Levine & Feizi, 2021) that completely nullify the need for sample correction under $\ell_1$ norm. Third, when we do need sample correction, working with binary variables allows us to use tighter concentration inequalities (Clopper & Pearson, 1934) (see § D.2 for a detailed discussion). Thus, even with significantly lower MC samples, our method still produces small prediction sets (see Fig. 1-left). This improvement is even more pronounced for datasets with a large number of classes (e.g. ImageNet shown in Fig. 5). Finally, BinCP does not require the score function to be bounded which is a limitation in current methods. Our code is available on the BinCP Github repository.

## 2 BACKGROUND

We assume a holdout set of labeled calibration datapoints $\mathcal{D}_{\text{cal}} = \{(\boldsymbol{x}_i, y_i)\}_{i=1}^n$ which is exchangeable with future test points $(\boldsymbol{x}_{n+1}, y_{n+1})$, both sampled from some distribution $\mathcal{D}$. We have blackbox access to a model from which we compute an arbitrary conformity[1] score $s : \mathcal{X} \times \mathcal{Y} \to \mathbb{R}$, e.g. score $s(\boldsymbol{x}, y) = \pi_y(\boldsymbol{x})$ where $\pi_y(\boldsymbol{x})$ is the predicted probability for class $y$ (other scores in § A).

**Vanilla CP.** For a user-specified nominal coverage $1 - \alpha$, let $q_\alpha = \mathbb{Q}(\alpha; \{s(\boldsymbol{x}_i, y_i)\}_{i=1}^n \cup \{\infty\})$ where $\mathbb{Q}(\cdot; \cdot)$ is the quantile function. The sets defined as $\mathcal{C}(\boldsymbol{x}_{n+1}) = \{y : s(\boldsymbol{x}_{n+1}, y) \geq q_\alpha\}$ have $1 - \alpha$ guarantee to include the true label $y_{n+1}$. Formally, $\Pr[y_{n+1} \in \mathcal{C}(\boldsymbol{x}_{n+1})] \geq 1 - \alpha$ (Vovk et al., 2005) where the probability is over $\mathcal{D}_{\text{cal}} \sim \mathcal{D}, \boldsymbol{x}_{n+1} \sim \mathcal{D}$. This guarantee, and later our robust sets, are independent of the mechanics of the model and the score function – the model's accuracy or the quality of the score function is irrelevant. A score function that better reflects input-label agreement leads to more efficient (i.e., smaller) prediction sets. For noisy or adversarial inputs, the

---

[1]Conformity scores quantify agreement and are equivalent up to a sign flip to non-conformity scores.

exchangeability between the test and calibration set breaks, making the coverage guarantee invalid. Fig. 1-middle, and Fig. 6-right show that an adversary (or bounded worst-case noise) can decrease the empirical coverage drastically with imperceptible perturbations on each test point. As a defense, *robust* CP extends this guarantee to the worst-case bounded perturbations.

**Threat model.** The adversary's goal is to decrease the empirical coverage probability by perturbing the input. Let $\mathcal{B} : \mathcal{X} \to 2^{\mathcal{X}}$ be a ball that returns all admissible perturbed points around an input. For images a common threat model is defined by the $\ell_2$ norm: $\mathcal{B}_r(\boldsymbol{x}) = \{\tilde{\boldsymbol{x}} : \|\tilde{\boldsymbol{x}} - \boldsymbol{x}\|_2 \leq r\}$ where the radius $r$ controls the perturbation magnitude. Similarly, we can use the $\ell_1$ norm. For binary data and graphs, Bojchevski et al. (2020) define $\mathcal{B}_{r_a,r_d}(\boldsymbol{x}) = \{\tilde{\boldsymbol{x}} : \sum_{i=1}^{d} \mathbb{I}[\tilde{\boldsymbol{x}}_i = \boldsymbol{x}_i - 1] \leq r_d, \sum_{i=1}^{d} \mathbb{I}[\tilde{\boldsymbol{x}}_i = \boldsymbol{x}_i + 1] \leq r_a\}$ where the adversary is allowed to toggle at most $r_a$ zero bits, and $r_d$ one bits.

**Inverted ball $\mathcal{B}^{-1}$.** At test time we are given a (potentially) perturbed $\tilde{\boldsymbol{x}} \in \mathcal{B}(\boldsymbol{x})$. However, to obtain robust sets, we need to reason about (the score) of the unseen clean $\boldsymbol{x}$. Naively, one might assume that $\boldsymbol{x} \in \mathcal{B}(\tilde{\boldsymbol{x}})$ – the clean point is in the ball around the perturbed point. However, this only holds in special cases such as the ball defined by the $l_2$ norm. For example, if a binary $\tilde{\boldsymbol{x}}$ was obtained by removing $r_d$ bits and adding $r_a$ bits, to able to reach the clean $\boldsymbol{x}$ from the perturbed $\tilde{\boldsymbol{x}}$ we need to add $r_d$ bits and remove $r_a$ bits instead since $\mathcal{B}_{r_a,r_d}$ unlike $\mathcal{B}_r$ is not symmetric. We define the inverted ball $\mathcal{B}^{-1}$ as the smallest ball centered at $\tilde{\boldsymbol{x}} \in \mathcal{B}(\boldsymbol{x})$ that includes the clean $\boldsymbol{x}$. Formally, $\mathcal{B}^{-1}$ should satisfy $\forall \tilde{\boldsymbol{x}} \in \mathcal{B}(\boldsymbol{x}) \Rightarrow \boldsymbol{x} \in \mathcal{B}^{-1}(\tilde{\boldsymbol{x}})$. For symmetric balls like $\ell_p$-norms, $\mathcal{B}^{-1} = \mathcal{B}$. For the binary ball $\mathcal{B}_{r_a,r_d}^{-1} = \mathcal{B}_{r_d,r_a}$ we need to swap $r_a$ and $r_d$ to ensure this condition. Zargarbashi et al. (2024) also discuss this subtle but important aspect without formally defining $\mathcal{B}^{-1}$.

**Robust CP.** Given a threat model, robust CP defines a *conservative* prediction set $\bar{\mathcal{C}}$ that maintains the conformal guarantee even for worst-case inputs. Formally,

$$\Pr_{\mathcal{D}_{\text{cal}} \cup \{\boldsymbol{x}_{n+1}\} \sim \mathcal{D}} [y_{n+1} \in \bar{\mathcal{C}}(\tilde{\boldsymbol{x}}_{n+1}), \forall \tilde{\boldsymbol{x}}_{n+1} \in \mathcal{B}(\boldsymbol{x}_{n+1})] \geq 1 - \alpha \tag{1}$$

The intuition behind existing methods is as follows: (i) Vanilla CP covers $\boldsymbol{x}_{n+1}$ with $1 - \alpha$ probability (ii) if $y \in \mathcal{C}(\boldsymbol{x}_{n+1})$ then $y \in \bar{\mathcal{C}}(\tilde{\boldsymbol{x}}_{n+1})$. Thus, robust CP covers $\tilde{\boldsymbol{x}}_{n+1}$ with at least the same probability. Here, (ii) is guaranteed via certified lower bounds $\text{c}^{\downarrow}[s, \boldsymbol{x}, \mathcal{B}]$ or certified upper bounds $\text{c}^{\uparrow}[s, \boldsymbol{x}, \mathcal{B}^{-1}]$.

**Theorem 1 (Robust CP from Zargarbashi et al. (2024)).** *Define $s_y(\cdot) = s(\cdot, y)$. With $\text{c}^{\uparrow}[s_y, \tilde{\boldsymbol{x}}, \mathcal{B}^{-1}] \geq \max_{\boldsymbol{x}' \in \mathcal{B}^{-1}(\tilde{\boldsymbol{x}})} s(\boldsymbol{x}', y)$, let $\bar{C}_{\text{test}}(\tilde{\boldsymbol{x}}_{n+1}) = \{y : \text{c}^{\uparrow}[s_y, \tilde{\boldsymbol{x}}_{n+1}, \mathcal{B}^{-1}] \geq q\}$, then $\bar{C}_{\text{test}}$ satisfies Eq. 1 (test-time robustness). Alternatively, with $\text{c}^{\downarrow}[s_y, \boldsymbol{x}, \mathcal{B}] \leq \min_{\boldsymbol{x}' \in \mathcal{B}(\boldsymbol{x})} s(\boldsymbol{x}', y)$, define $q^{\downarrow} = \mathbb{Q}\left(\alpha; \{\text{c}^{\downarrow}[s_{y_i}, \boldsymbol{x}_i, \mathcal{B}]\}_{i=1}^{n}\right)$. Then $\bar{C}_{\text{cal}}(\tilde{\boldsymbol{x}}_{n+1}) = \{y : s(\tilde{\boldsymbol{x}}_{n+1}, y) \geq q^{\downarrow}\}$ also satisfies Eq. 1 (calibration-time robustness).*

In Theorem 1 test-time robustness uses $\mathcal{B}^{-1}$ since it queries the clean point from the perspective of the perturbed test input. Alternatively, calibration-time robustness uses $\mathcal{B}$ since the clean calibration point is given and we are finding the lower bound for the unseen test point in the test. The intuition is that the lower bound scores from the clean calibration points are exchangeable with the lower bound of the clean test input. The perturbed test input will surely have a higher score compared to this lower bound, hence it would be covered with higher probability.

We can obtain the $\text{c}^{\downarrow}, \text{c}^{\uparrow}$ bounds through neural network verifiers Jeary et al. (2024) or randomized smoothing (Cohen et al., 2019). We focus on the latter since we get model-agnostic certificates with black-box access. The coverage probability is theoretically proved in CP. Similarly, (adversarially) robust CP also comes with a theoretical guarantee. In both cases we can compute the empirical coverage as a sanity check. Another metric of interest in both cases is the average set size (the efficiency) of the conformal sets.

**Randomized smoothing.** A smoothing scheme $\xi : \mathcal{X} \to \mathcal{X}$ maps any point to a random nearby point. For continuous data Gaussian smoothing $\xi(\boldsymbol{x}) = \boldsymbol{x} + \boldsymbol{\epsilon}$ adds an isotropic Gaussian noise to the input $\boldsymbol{\epsilon} \sim \mathcal{N}(\boldsymbol{0}, \sigma\boldsymbol{I})$. For sparse binary data Bojchevski et al. (2020) define sparse smoothing as $\xi(\boldsymbol{x}) = \boldsymbol{x} \oplus \boldsymbol{\epsilon}$ where $\oplus$ is the binary XOR, and $\boldsymbol{\epsilon}[i] \sim \text{Bernoulli}(p = p_{\boldsymbol{x}[i]})$, where $p_1$, and $p_0$ are two smoothing parameters to account for sparsity. To simplify the notation we write $\boldsymbol{x} + \boldsymbol{\epsilon}$ instead of $\xi(\boldsymbol{x})$ in the rest of the paper for both Gaussian and sparse smoothing, but our method works for any smoothing scheme beyond additive noise. Regardless of how rapidly a score function $s(\boldsymbol{x}, y)$ changes, the smooth score $\bar{s}(\boldsymbol{x}, y) = \mathbb{E}_{\boldsymbol{\epsilon}}[s(\boldsymbol{x} + \boldsymbol{\epsilon}, y)]$ changes slowly near $\boldsymbol{x}$. This enables us to compute tight $\text{c}^{\downarrow}, \text{c}^{\uparrow}$ bounds that depend on the smoothing strength. See § 4, § B, and § D.1 for details.

## 3 BINARIZED CONFORMAL PREDICTION (BINCP)

We define conformal sets by binarizing randomized scores. We first show that this preserves the conformal guarantee for clean data. Then in § 4 we extends the guarantee to worst-case adversarial inputs. As we will see in § 6 our binarization approach has gains in terms of Monte-Carlo sampling budget, computational cost, and average set size.

**Proposition 1.** *For any two parameters $p \in (0,1), \tau \in \mathbb{R}$, given a smoothing scheme $\boldsymbol{x} + \boldsymbol{\epsilon}$, define the boolean function $\text{accept}[\cdot, \cdot; p, \tau]$ and the prediction set $\mathcal{C}(\cdot; p, \tau)$ as*

$$\text{accept}[\boldsymbol{x}, y; p, \tau] = \mathbb{I}[\Pr_{\boldsymbol{\epsilon}}[s(\boldsymbol{x} + \boldsymbol{\epsilon}, y) \geq \tau] \geq p] \quad and \quad \mathcal{C}(\boldsymbol{x}; p, \tau) = \{y : \text{accept}(\boldsymbol{x}, y; p, \tau)\}$$

*For any fixed $p$, let*

$$\tau_\alpha(p) = \sup_\tau \left\{ \tau : \sum_{i=1}^n \text{accept}(\boldsymbol{x}_i, y_i; p, \tau) \geq (1 - \alpha) \cdot (n + 1) \right\} \tag{2}$$

*then the set $\mathcal{C}(\boldsymbol{x}_{n+1}; p, \tau_\alpha(p))$ has $1 - \alpha$ coverage guarantee. Alternatively, for any fixed $\tau$, let*

$$p_\alpha(\tau) = \sup_p \left\{ p : \sum_{i=1}^n \text{accept}(\boldsymbol{x}_i, y_i; p, \tau) \geq (1 - \alpha) \cdot (n + 1) \right\} \tag{3}$$

*again the prediction set $\mathcal{C}(\boldsymbol{x}_{n+1}; p_\alpha(\tau), \tau)$ has $1 - \alpha$ coverage guarantee.*

The correctness of Prop. 1 can be directly seen by noticing that we implicitly define new scores.

**Quantile view.** Let $S_i = s(\boldsymbol{x}_i + \boldsymbol{\epsilon}, y_i)$ be the distribution of randomized scores for $\boldsymbol{x}_i$ and the true class $y_i$. Let $\tau_i(p) = \mathbb{Q}(p; S_i)$, we have that $\tau_\alpha(p) = \mathbb{Q}(\alpha; \{\tau_i(p)\}_{i=1}^n)$ is a quantile of quantiles. Similarly, define $p_i(\tau) = \mathbb{Q}^{-1}(\tau; S_i)$ then $p_\alpha(\tau) = \mathbb{Q}(\alpha; \{p_i(\tau)\}_{i=1}^n)$ is a quantile of inverse quantiles. Both $\tau_i(p)$ for a fixed $p$ and $p_i(\tau)$ for a fixed $\tau$ are valid conformity scores for the instance $\boldsymbol{x}_i$, since exchangeability is trivially preserved. Therefore, $\tau_\alpha(p)$ and $p_\alpha(\tau)$ are just the standard quantile thresholds from CP on some new score functions. This directly gives the $1 - \alpha$ coverage guarantee. This view via the implicit scores is helpful for intuition, but we keep the original formulation since it is more directly amenable to certification as we show in § 4. We provide an additional formal proof of Prop. 1 via conformal risk control (Angelopoulos et al., 2022) in § C.

Using either variant from Prop. 1 let $(p_\alpha, \tau_\alpha)$ equal $(p, \tau_\alpha(p))$ or $(p_\alpha(\tau), \tau)$ as the final pair of parameters. For test points $\boldsymbol{x}_{n+1}$ we accept labels whose smooth score distribution has at least $p_\alpha$ proportion above the threshold $\tau_\alpha$, i.e. $\text{accept}(\boldsymbol{x}_{n+1}, y; p_\alpha, \tau_\alpha) = 1$. The term "binarization" refers to mapping each score sample above $\tau$ to 1 and all others 0. For distributions with a strictly increasing and continuous CDF (e.g. isotropic Gaussian smoothing) both variants are equivalent.

**Lemma 1.** *Given distributions $\{S_i\}_{i=1}^n$ with strictly increasing and continuous CDFs, let $\tau_\alpha(p)$ be obtained from Eq. 2 with fixed $p$ and $p_\alpha(\cdot)$ be as defined in Eq. 3. We have $p_\alpha(\tau_\alpha(p)) = p$.*

We defer all proofs to § C. For fixed $p$, Prop. 1 yields $(p, \tau_\alpha(p))$. Fixing $\tau = \tau_\alpha(p)$ we get sets with $(p_\alpha(\tau), \tau) = (p_\alpha(\tau_\alpha(p)), \tau_\alpha(p))$ which also equals $(p, \tau_\alpha(p))$ from Lemma 1. Fig. 2 shows the $\text{accept}(\boldsymbol{x}, y; p, \tau)$ function for several examples. This function is non-increasing in both parameters

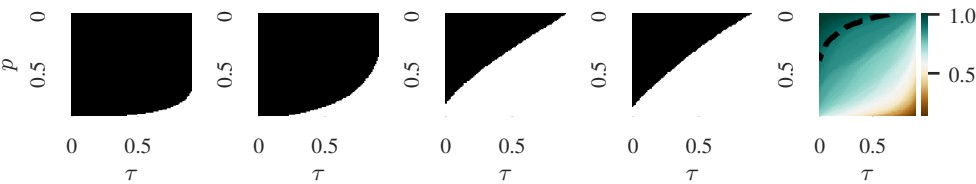

Figure 2: [Left] Function $\text{accept}(\boldsymbol{x}_i, y_i; p, \tau)$ for different $(p, \tau)$ pairs for four random CIFAR-10 instances. Black equals 1 and white equals 0. [Right] Empirical coverage for different $(p, \tau)$ pairs. Any $(p, \tau)$ pair on the dashed black line (the 0.9 contour) gives conformal sets with 90% coverage.

$p$ and $\tau$. In general, any arbitrary assignment of $p$, and $\tau$, results in some expected coverage – $\mathrm{accept}(\cdot, \cdot, p, \tau)$ equals to 1 for some number of $(\boldsymbol{x}_i, y_i)$s (Fig. 2-right). Pairs $(p_\alpha, \tau_\alpha)$ obtained from Prop. 1 are placed on the $1 - \alpha$ contour of this expectation. The empirical coverage is close to this expectation due to exchangeability (Berti & Rigo, 1997).

**Remarks.** The scores $\tau_i(p)$ (and similarly $p_i(\tau)$ remain exchangeable whether the quantile over the smoothing distribution is computed exactly or estimated from any number of Monte-Carlo samples. That is, Prop. 1 holds regardless. However, when need to be more careful when we consider the certified upper and lower bounds. In § 4 we first derive robust conservative sets that maintain worst-case coverage, assuming that we can compute probabilities and expectations exactly. Since this is not always possible, in § 5 we provide the appropriate sample correction that still preserves the robustness guarantee when using Monte-Carlo samples. We also discuss a de-randomized approach that does not need sample correction.

## 4 ROBUST BINCP

From Prop. 1 (either variant) we compute a pair $(p_\alpha, \tau_\alpha)$. Following Prop. 1, for clean $\boldsymbol{x}_{n+1}$, we have $\Pr[s(\boldsymbol{x}_{n+1} + \boldsymbol{\epsilon}, y_{n+1}) \geq \tau_\alpha] \geq p_\alpha$ with probability $1 - \alpha$. We will exploit this property. Define $f_y(\boldsymbol{x}) = \mathbb{I}[s(\boldsymbol{x}, y) \geq \tau_\alpha]$, we have $\bar{f}_y(\boldsymbol{x}) = \mathbb{E}_{\boldsymbol{\epsilon}}[\mathbb{I}[s(\boldsymbol{x} + \boldsymbol{\epsilon}, y) \geq \tau_\alpha]] = \Pr_{\boldsymbol{\epsilon}}[s(\boldsymbol{x} + \boldsymbol{\epsilon}, y) \geq \tau_\alpha]$.

**Conventional robust CP.** One way to attain robust prediction sets is to apply the same recipe as Zargarbashi et al. (2024) (CAS) by finding upper or lower bounds on the new score function. CAS uses the smooth score $\bar{s}_y(\boldsymbol{x}) = \mathbb{E}_{\boldsymbol{\epsilon}}[s(\boldsymbol{x} + \boldsymbol{\epsilon}, y)]$. Instead, we can bound $\bar{f}_y(\boldsymbol{x})$ which is a smooth binary classifier. Note that as discussed in § 3 (quantile of inverse quantiles), $\bar{f}_y(\boldsymbol{x})$ is a conformity score function itself. Therefore, following Theorem 1, the test-time, and calibration-time robust prediction sets are

$$\bar{\mathcal{C}}_{\text{test}}(\tilde{\boldsymbol{x}}_{n+1}) = \{y : \mathrm{c}^\uparrow[\bar{f}_y, \tilde{\boldsymbol{x}}_{n+1}, \mathcal{B}^{-1}] \geq p_\alpha\}, \quad \bar{\mathcal{C}}_{\text{cal}}(\tilde{\boldsymbol{x}}_{n+1}) = \{y : \bar{f}_y(\tilde{\boldsymbol{x}}_{n+1}) \geq q^\downarrow\} \quad (4)$$

where $q^\downarrow = \mathbb{Q}\left(\alpha; \{\mathrm{c}^\downarrow[\bar{f}_{y_i}, \boldsymbol{x}_i, \mathcal{B}]\}_{i=1}^n\right)$. In short, we replace the clean $\bar{f}_{y_{n+1}}(\boldsymbol{x}_{n+1})$ with either its certified upper $\mathrm{c}^\uparrow$ or lower $\mathrm{c}^\downarrow$ bound. We elaborate on this approach before improving it.

**Computing $\mathrm{c}^\downarrow$ and $\mathrm{c}^\uparrow$.** Computing exact worst-case bounds on $\bar{f}$ ($\bar{f}_y$ for all $y$) is intractable and requires white-box access to the score function and therefore the model. Following established techniques in the randomized smoothing literature (Lee et al., 2019) we relax the problem. Formally,

$$\mathrm{c}^\downarrow[\bar{f}, \boldsymbol{x}, \mathcal{B}] = \min_{\substack{\tilde{\boldsymbol{x}} \in \mathcal{B}(\boldsymbol{x}) \\ h \in \mathcal{H}}} \Pr_{\boldsymbol{\epsilon}}[h(\tilde{\boldsymbol{x}} + \boldsymbol{\epsilon})] \quad \text{s.t.} \quad \Pr_{\boldsymbol{\epsilon}}[h(\boldsymbol{x} + \boldsymbol{\epsilon})] = \Pr_{\boldsymbol{\epsilon}}[f(\boldsymbol{x} + \boldsymbol{\epsilon})] = \bar{f}(\boldsymbol{x}) \quad (5)$$

where $\mathcal{H}$ is the set of all measurable functions $h$. Since $f \in \mathcal{H}$ we have $\mathrm{c}^\downarrow[\bar{f}, \boldsymbol{x}, \mathcal{B}] \leq \bar{f}(\tilde{\boldsymbol{x}})$ for all $\tilde{\boldsymbol{x}} \in \mathcal{B}(\boldsymbol{x})$. The upper bound $\mathrm{c}^\uparrow[\bar{f}, \boldsymbol{x}, \mathcal{B}^{-1}]$ is the solution to a similar *maximization* problem.

**Closed form.** For $\ell_2$ ball with Gaussian smoothing, Eq. 5 has a closed form solution $\Phi_\sigma(\Phi_\sigma^{-1}(\bar{f}_y(\boldsymbol{x})) - r)$ where $\Phi_\sigma$ is the CDF of the normal distribution $\mathcal{N}(\boldsymbol{0}, \sigma \boldsymbol{I})$(Cohen et al., 2019; Kumar et al., 2020). The upper bound is similarly computed by changing the sign of $r$. Yang et al. (2020) show the same closed-form applies solution for the $\ell_1$ ball, and additionally, discuss other perturbation balls and smoothing schemes most of which are applicable. For sparse smoothing we can compute the bounds with a simple algorithm with $O(r_a + r_d)$ runtime (Bojchevski et al., 2020), which we discuss in § C. For $\ell_1$ ball and uniform smoothing the lower bound equals $\bar{f}_y(\boldsymbol{x}) - 1/(2\lambda)$ where $\boldsymbol{\epsilon} \sim \mathcal{U}[0, 2\lambda]^d$ (Levine & Feizi, 2021). This bound can also be de-randomized (see § 5).

**Single Binary Certificate.** From the closed-form solutions we see that the bounds are independent of the definition of $f$, and the test point $\boldsymbol{x}$; i.e. their output is a function of the scalar $p := \bar{f}_y(\boldsymbol{x})$. We defer the discussion for why this holds to § B, and § D.1; in short the solution for any $\boldsymbol{x}$ can be obtained from alternative canonical points $\boldsymbol{u}$, and $\tilde{\boldsymbol{u}}$. Therefore, we write $\mathrm{c}^\downarrow[p, \mathcal{B}] = \mathrm{c}^\downarrow[\bar{f}_y, \boldsymbol{x}, \mathcal{B}]$ to show that $\mathrm{c}^\downarrow$ depends only on $p$ and $\mathcal{B}$, and the same for $\mathrm{c}^\uparrow$. We also notice that in common smoothing schemes and perturbation balls, it holds that $\mathrm{c}^\downarrow[\mathrm{c}^\uparrow[p, \mathcal{B}^{-1}], \mathcal{B}] = p$ which allows us to reduce both calibration-time and test-time robustness to solving a single binary certificate. We formalize this in Lemma 2.

**Lemma 2.** *If* $\mathrm{c}^\downarrow[\mathrm{c}^\uparrow[p, \mathcal{B}^{-1}], \mathcal{B}] = p$ *for all $p$, then* $\bar{\mathcal{C}}_{\text{test}}(\tilde{\boldsymbol{x}}_{n+1}) = \bar{\mathcal{C}}_{\text{cal}}(\tilde{\boldsymbol{x}}_{n+1}) = \bar{\mathcal{C}}_{\text{bin}}(\tilde{\boldsymbol{x}}_{n+1})$ *where* $\bar{\mathcal{C}}_{\text{bin}}(\tilde{\boldsymbol{x}}_{n+1}) = \{y : \mathrm{accept}(\tilde{\boldsymbol{x}}_{n+1}, y; \mathrm{c}^\downarrow[p_\alpha, \mathcal{B}], \tau_\alpha)\} = \{y : \Pr_{\boldsymbol{\epsilon}}[s(\boldsymbol{x}_{n+1} + \boldsymbol{\epsilon}, y_{n+1}) \geq \tau_\alpha] \geq \mathrm{c}^\downarrow[p_\alpha, \mathcal{B}]\}$.

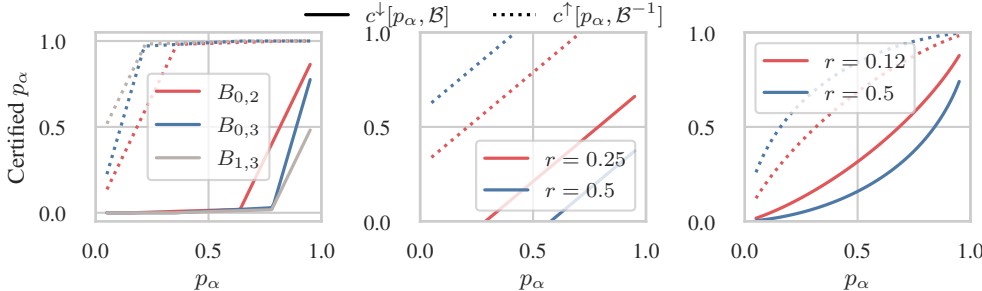

Figure 3: [From left to right] Certified bounds for sparse smoothing, $\ell_1$ ball with de-randomized uniform smoothing (Levine & Feizi, 2021), and $\ell_2$ (same as $\ell_1$) ball with Gaussian smoothing.

To see why, let $\tilde{p}_{n+1} = \bar{f}_{y_{n+1}}(\tilde{\boldsymbol{x}}_{n+1})$. The test-time robust coverage requires $c^{\uparrow}[\tilde{p}_{n+1}, \mathcal{B}^{-1}] \geq p_{\alpha}$. Since both $c^{\downarrow}$, and $c^{\uparrow}$ are non-decreasing w.r.t. $p$, we have $c^{\downarrow}[c^{\uparrow}[\tilde{p}_{n+1}, \mathcal{B}^{-1}], \mathcal{B}] \geq c^{\downarrow}[p_{\alpha}, \mathcal{B}]$. We have the equivalent condition $\tilde{p}_{n+1} \geq c^{\downarrow}[p_{\alpha}, \mathcal{B}]$. This implies that we only need to compute a single certificate $c^{\downarrow}[p_{\alpha}, \mathcal{B}]$ once with the single $p_{\alpha}$ value given by Prop. 1. This also allows us to seamlessly integrate other existing binary certificates in a plug and play manner. In contrast, with Theorem 1 for $\bar{\mathcal{C}}_{\text{test}}$ or $\bar{\mathcal{C}}_{\text{cal}}$ we need at least one certificate per test (or calibration) point. Notably, these prediction set are identical to our cheaper $\bar{\mathcal{C}}_{\text{bin}}$. For illustration, Fig. 3 shows the certified lower and upper bounds for various $p_{\alpha}$ values and various smoothing schemes.

Intuitively $c^{\downarrow}[c^{\uparrow}[p, \mathcal{B}^{-1}], \mathcal{B}] = p$ holds due to symmetry of the smoothing scheme w.r.t. $\mathcal{B}$, and $\mathcal{B}^{-1}$ and is satisfied by most smoothing schemes. In Lemma 3 we prove that Gaussian, uniform, and sparse smoothing all have this property.

**Lemma 3.** *For Gaussian, and uniform smoothing under $\ell_1$, and $\ell_2$ balls $\mathcal{B}_r = \mathcal{B}_r^{-1}$. For sparse smoothing and $\mathcal{B}_{r_a, r_d}$ we have $\mathcal{B}_{r_a, r_d}^{-1} = \mathcal{B}_{r_d, r_a}$. In all three cases we have $c^{\downarrow}[c^{\uparrow}[p, \mathcal{B}^{-1}], \mathcal{B}] = p$.*

To summarize, for robust BinCP, we first compute conformal thresholds $(p_{\alpha}, \tau_{\alpha})$ from Prop. 1. Then for a perturbation ball $\mathcal{B}$ that satisfies $c^{\downarrow}[c^{\uparrow}[p, \mathcal{B}^{-1}], \mathcal{B}] = p$, we compute $c^{\downarrow}[p_{\alpha}, \mathcal{B}]$ and compute the prediction sets with $(c^{\downarrow}[p_{\alpha}, \mathcal{B}], \tau_{\alpha})$ instead. The resulting sets have $1 - \alpha$ robust coverage.

**Corollary 1.** *With $(p_{\alpha}, \tau_{\alpha})$ from Prop. 1 on a calibration set $\mathcal{D}_{\text{cal}}$, let $\boldsymbol{x}_{n+1}$ be exchangeable with $\mathcal{D}_{\text{cal}}$ and $\tilde{\boldsymbol{x}}_{n+1} \in \mathcal{B}(\boldsymbol{x}_{n+1})$. If for the smoothing scheme $\xi$ and the threat model $\mathcal{B}$ and for all $p$ we have $c^{\downarrow}[c^{\uparrow}[p, \mathcal{B}^{-1}], \mathcal{B}] = p$, then the set $\bar{\mathcal{C}}_{\text{bin}}(\tilde{\boldsymbol{x}}_{n+1}) = \{y : \Pr[s(\tilde{\boldsymbol{x}}_{n+1} + \boldsymbol{\epsilon}, y_{n+1}) \geq \tau_{\alpha}] \geq c^{\downarrow}[p_{\alpha}, \mathcal{B}]\}$ has $1 - \alpha$ coverage (the pseudocode is in § A).*

## 5 ROBUST BINCP WITH FINITE SAMPLES

The certificate in Corollary 1 relies on exact probabilities $\Pr[s(\boldsymbol{x}_{n+1} + \boldsymbol{\epsilon}, y_{n+1}) \geq \tau_{\alpha}]$ which is often intractable to compute. Instead, we can either apply de-randomization techniques or estimate high-confidence bounds of these probabilities. We first describe the latter approach. For each calibration point $(\boldsymbol{x}_i, y_i)$ we compute $q_i = \frac{1}{m} \sum_{i=1}^{m} \mathbb{I}[s(\boldsymbol{x}_i + \boldsymbol{\epsilon}, y_i) \geq \tau_{\alpha}]$ where $m$ is the number of Monte-Carlo (MC) samples. For each label of the (potentially perturbed) test point we compute $\tilde{q}_{n+1,y} = \frac{1}{m} \sum_{i=1}^{m} \mathbb{I}[s(\tilde{\boldsymbol{x}}_{n+1} + \boldsymbol{\epsilon}, y) \geq \tau_{\alpha}]$. We use the Clopper-Pearson confidence interval (Clopper & Pearson, 1934) to bound the exact probabilities via the MC estimates. To ensure the sets are conservative we compute a lower bound for calibration points and an upper bound for test points. Collectively, all bounds are valid with adjustable $1 - \eta$ probability. To account for this, we set the nominal coverage level to $1 - \alpha + \eta$ such that we have $1 - \alpha$ coverage in total. Similar to Zargarbashi et al. (2024), we compute each bound with $1 - \eta/(|\mathcal{D}_{\text{cal}} + k|)$ probability where $k$ is the number of classes. Let $p_i = \Pr[s(\boldsymbol{x}_i + \boldsymbol{\epsilon}, y_i) \geq \tau_{\alpha}]$ for $i \in \{1, \ldots, n + 1\}$ be the exact probabilities. The final sample-corrected robust predictions sets are given in Prop. 2.

**Proposition 2.** *Let $q_i^{\downarrow} \leq p_i$ hold with $1 - \eta/(\mathcal{D}_{\text{cal}} + k)$ for each calibration point $i \in \{1, \ldots, n\}$ where $k$ is the number of target classes. For a given test point $\tilde{\boldsymbol{x}}_{n+1}$ let $\tilde{q}_{n+1,y}^{\uparrow} \geq \tilde{p}_{n+1,y}$ with $1 - \eta/(\mathcal{D}_{\text{cal}} + k)$ where $\tilde{p}_{n+1,y} = \Pr[s(\tilde{\boldsymbol{x}}_{n+1} + \boldsymbol{\epsilon}, y) \geq \tau_{\alpha}]$. With $p_{\alpha}^{\downarrow} = \mathbb{Q}\left(\alpha - \eta; \{q_i^{\downarrow}\}_{i=1}^{n}\right)$,*

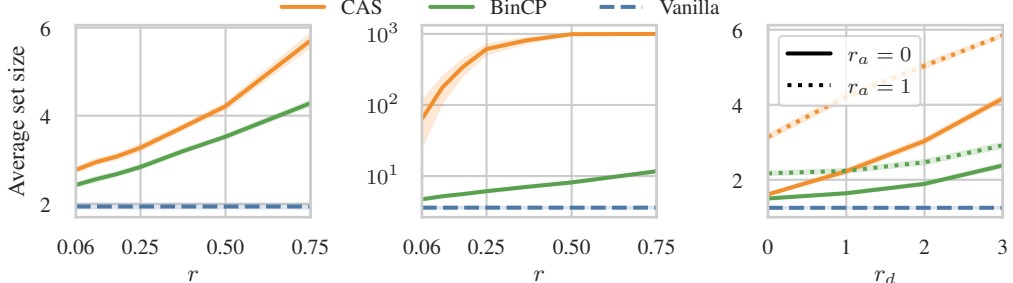

Figure 4: [Left to right] Average prediction set size of robust CP for CIFAR-10, and ImageNet with Gaussian smoothing ($\sigma = 0.5$), and CoraML with sparse smoothing. All results are for 2000 Monte-Carlo samples. We set $1 - \alpha = 0.85$ for ImageNet, and $1 - \alpha = 0.9$ for CIFAR-10 and CoraML.

*we set the robust conformal threshold pair as* $(\mathrm{c}^{\downarrow}[p_\alpha^{\downarrow}, \mathcal{B}], \tau_\alpha)$. *Then the prediction set defined as* $\bar{\mathcal{C}}_+(\tilde{x}_{n+1}; \mathrm{c}^{\downarrow}[p_\alpha^{\downarrow}, \mathcal{B}], \tau_\alpha) = \{y : \tilde{q}_{n+1,y}^{\uparrow} \geq \mathrm{c}^{\downarrow}[p_\alpha^{\downarrow}, \mathcal{B}]\}$ *has* $1 - \alpha$ *coverage probability.*

Such sample correction is a crucial step for smoothing-based robust CP, since the robustness certificate is probabilistic. The failure of the certificate depends to the failure of the confidence intervals. In contrast, for deterministic and de-randomized certificates such as DSSN (Levine & Feizi, 2021), we do not need sample correction since we can exactly compute $p_i$ and $p_\alpha = \mathbb{Q}(\alpha; \{p_i\}_{i=1}^n)$. Note, vanilla (non-robust) BinCP does not need sample correction to maintain the guarantee (see § 3).

## 6 EXPERIMENTS

We show that: (i) We can return guaranteed and small sets for both image classification and node classification, with a significantly lower number of Monte Carlo samples. (ii) Our sets are computationally efficient. (iii) There is an inherent robustness in randomized methods. (iv) We can also use de-randomized smoothing-based certificates that do not require finite sample correction.

**Setup.** We evaluate our method on two image datasets: CIFAR-10 (Krizhevsky, 2009) and ImageNet (Deng et al., 2009), and for node-classification (graph) dataset we use Cora-ML McCallum et al. (2004). For the CIFAR-10 dataset we use `ResNet-110` and for the ImageNet dataset we use ResNet-50 pretrained models with noisy data augmentation from Cohen et al. (2019). For the graph classification task we similarly train a GCN model Kipf & Welling (2017) on CoraML with noise augmentation. The GCN is trained with 20 nodes per class with stratified sampling as the training set (and similarly sampled validation set). The size of the calibration set is between 100 and 250 (sparsely labeled setting) unless specified explicitly. Our reported results on conformal prediction performance are averaged over 100 runs with different calibration set samples. We calibrated BinCP with a $p = 0.6$ fixed value, however small changes in $p$ do not influence the result. For the graph dataset we calibrated BinCP with $p = 0.9$. Intuitively as $\mathrm{c}^{\downarrow}[p, \mathcal{B}]$ has a sharp decay for sparse smoothing (see Fig. 10-left), we set $p$ to a number such that $\mathrm{c}^{\downarrow}[p, \mathcal{B}]$ still remains high.

We conducted our experiment using three different smoothing schemes. (i) Smoothing with isotropic Gaussian noise, $\sigma = 0.12, 0.25$, and $0.15$. Our reported results for BinCP are valid for both $\ell_1$, and $\ell_2$ perturbation balls. (ii) De-randomized smoothing with splitting noise (DSSN) from Levine & Feizi (2021) from which we attain $\ell_1$ robustness. We examine two smoothing levels $\lambda = 0.25/\sqrt{3}$, and $0.5/\sqrt{3}$. (iii) Sparse smoothing from Bojchevski et al. (2020) with $p_+ = 0.01$, and $p_- = 0.6$ on node attributes. We report robustness across $r_a \in \{0, 1\}$, and $r_d \in \{0, 1, 2, 3\}$. We compare our the result from BinCP to the SOTA method CAS (Zargarbashi et al., 2024). Previously it was shown that CAS significantly outperforms RSCP (Gendler et al., 2021) both with and without finite sample correction. In § 7 we discuss the other related works in detail. In the standard setup, we estimate the statistics (mean and CDF, or Bernoulli parameters) with $2 \times 10^3$ Monte-Carlo samples, and we set $1 - \alpha = 0.9$. This setup is picked in favor of the baseline since by increasing the nominal coverage or decreasing the sample size BinCP outperforms the baseline with an even higher margin. Throughout the paper we report different nominal coverages and MC sampling budgets.

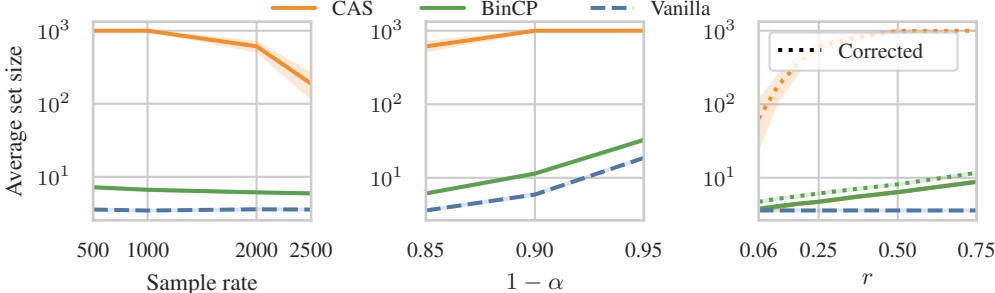

Figure 5: On ImageNet dataset, [left] average set size for $1 - \alpha = 0.85$ with various MC sampling budgets. [Middle] Set size across various levels of $1 - \alpha$ for $2 \times 10^3$ samples. [Right] Set size without sample correction (asymptotically valid assumption). The sample-corrected variants are shown with a dotted line. In all plots $y$-axis is log-scaled.

**Smaller set size.** Fig. 4 shows that for all datasets, and both smoothing schemes (isotropic Gaussian and sparse smoothing), BinCP produces smaller prediction sets compared to CAS. Our prediction sets computed with Gaussian noise are robust to both $\ell_1$, and $\ell_2$ perturbation balls with the same radius (Yang et al., 2020). In Fig. 5, compare to CAS on ImageNet dataset over various sample rates, coverages, and radii. Since CAS with sample correction returns trivial sets for $1 - \alpha = 0.9$ (see Fig. 5-middle), in Fig. 5-left we compare the results for $1 - \alpha = 0.85$. By increasing the Monte Carlo sample rate, the average set sizes of CAS and BinCP become closer – Fig. 1-left for CIFAR-10, Fig. 5-left for ImageNet, and Fig. 7-left for CoraML depict the impact of higher sampling budget. Intuitively as the number of classes increase (e.g. ImageNet dataset) the union bound over classes result in larger prediction sets (looser confidence intervals). In § E (Fig. 9) we show that BinCP is also consistently better for smaller radii (a setup similar to verifier based robust CP (Jeary et al., 2024)). For the same reported set size, we attain robust CP of significantly larger radii.

Note that Fig. 4-right, and Fig. 7-left show the performance on the *transductive* node classification setting where perfect robustness is achievable for free through memorization (see discussion by Gosch et al. (2023)). Nonetheless, comparing BinCP to CAS is still meaningful. Constructing robust conformal sets for GNNs in a realistic (inductive) setup is more challenging as discussed in § D.3.

**Exact $\ell_1$ robustness.** Using the de-randomized DSSN certificate for $\ell_1$ perturbation ball (Levine & Feizi, 2021) we derive the first smoothing-based de-randomized robust CP. As shown in Fig. 6-middle the de-randomized robust BinCP with uniform noise results in a significantly smaller set size across all radii compared to Gaussian noise (Fig. 4-left). Notably due to exactness of the computed statistics, for randomized DSSN-based certificate we bypass the finite samples correction.

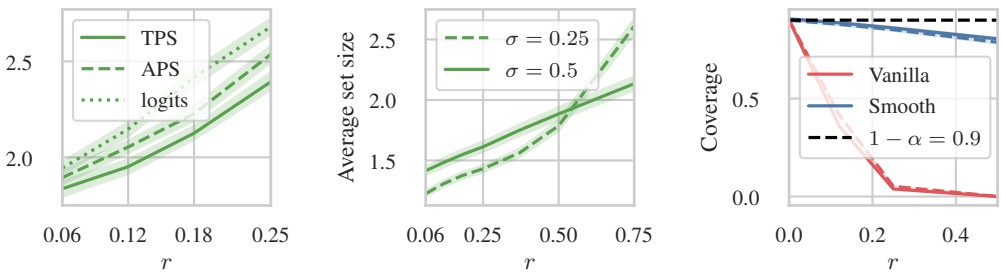

Figure 6: [Left] Performance of BinCP on different score functions under Gaussian smoothing with $\sigma = 0.25$. [Middle] Set size of BinCP with $\ell_1$ robustness and derandomized DSSN smoothing ($\sigma = \lambda/\sqrt{3}$). [Right] Vanilla non-smooth and smooth ($\sigma = 0.25$) prediction (solid and dashed colored lines show TPS and APS score function) under attack. All results are on CIFAR-10.

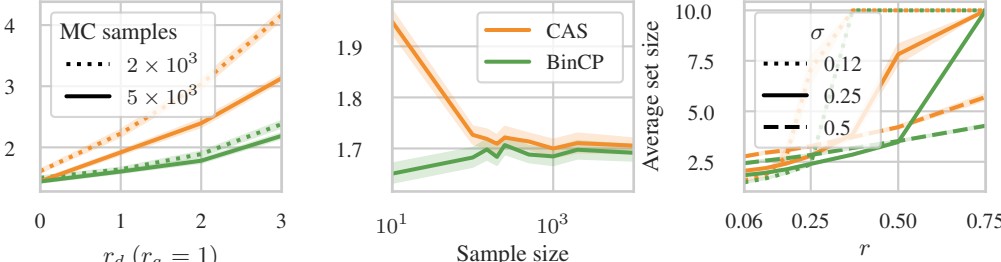

Figure 7: Comparison between BinCP and CAS for [Left] the effect of higher MC sample budget in CoraML dataset (sparse smoothing), [Middle] effect of low samples without finite samples correction for CIFAR-10 dataset ($\sigma = 0.25$), and [Right] various smoothing strengths $\sigma$.

**Ignoring sample correction.** While unrealistic in practice, Gendler et al. (2021) report results without applying finite sample correction. Zargarbashi et al. (2024) maintain small set sizes (with large MC sample rate) for CIFAR-10. However, for ImageNet and CoraML they only report the results without correction, and therefore with an "asymptomatically valid" coverage guarantee – valid when sample rate approaches infinity. In Fig. 5-left applying sample correction to CAS on datasets like ImageNet, inflates the prediction sets up to $\mathcal{Y}$, likely due to union bound on a large number of classes. Fig. 5-right shows that on ImageNet, both methods show similar prediction sets for asymptotically valid setup. Notably BinCP with sample correction is not far from the non-corrected setup, while CAS shows a large gap. Similarly Fig. 7-left shows that for sparse smoothing, increasing sample rate helps CAS considerably while its effect on BinCP is almost negligible. In § D.2 we explore the intuition behind how binarization can mitigate the impact of finite sample correction with fewer samples.

**Number of samples.** The upperbound in CAS is obtained through a two step process. First given the corrected CDF, we compute the worst case (adversarial) CDF. Then using upper bounded (or lower bounded) CDF, we apply the Anderson bound to obtain a bound on the mean from the CDF (Zargarbashi et al., 2024). Increasing the number of bins increases the computation slightly but produces tighter bounds. To observe this effect, without sample correction, we decrease the number of samples to a very low number ($\sim 10$, however unrealistic) and in Fig. 7-middle we see that set size in CAS slightly increases even without accounting for finite sample estimation.

**Effect of $\sigma$, and score function.** The strength of Gaussian smoothing is controlled by $\sigma$ in $\xi(\boldsymbol{x}) = \boldsymbol{x} + \boldsymbol{\epsilon}, \boldsymbol{\epsilon} \sim \mathcal{N}(\boldsymbol{0}, \sigma^2 \boldsymbol{I})$. In Fig. 7-right we observe a trade-off in choosing $\sigma$ for both methods. Higher smoothing intensity results in larger set sizes in the beginning, but by increasing the robustness radius the set size increases slowly. Still in all cases BinCP outperforms CAS. It is best practice to compute smooth prediction probabilities using a model trained with similar noise augmentation. We reported this result in § E (Table 2). Interestingly, when training and inference $\sigma$ does not match, BinCP shows good results while CAS is more sensitive.

We mainly focus on TPS score function, in addition Fig. 6-left compares APS, and logit as score functions. Here across all radii TPS is more efficient. We further report results for APS in § E. Since logits are unbounded, CAS is not applicable to it.

**Benefits of smoothing.** The guarantee of robust CP breaks for adversarial (or noisy) inputs. In Fig. 6-right we compare vanilla prediction and smooth prediction sets under adversarial attack. Notably, smooth models even without a conservative certificate show an inherent robustness. As illustrated, the non-smooth model quickly breaks to near 0 coverage guarantee for very small $r$. Relatedly, recent verifier-based robust CP (Jeary et al., 2024) report comparably larger prediction sets even for one order of magnitude smaller radius (compared to the certified radii by BinCP). This intuitively suggests that for robust CP it seems that randomization is inherently beneficial.

**Limitations.** While BinCP reduce the required MC sample rate significantly (e.g. from 2000 to 150 on CIFAR-10), still this number of inferences is computationally intensive. The robustness in the input space $\mathcal{X}$ is not yet linked to robustness w.r.t. distribution shift. Although BinCP applies on sparse smoothing, a realistic threat model for graphs (inductive GNNs) is still not addressed.

## 7 RELATED WORK

**Robust CP via smoothing.** Gendler et al. (2021) introduced the problem and defined a baseline robust CP method, RSCP (randomly smoothed conformal prediction), which applies Theorem 1 in combination with the mean-constrained upper bound for $\ell_2$ perturbations and Gaussian smoothing. This upper bound has a closed form solution: $\overline{s}(\tilde{\boldsymbol{x}}, y) = \Phi(\Phi^{-1}(p) + r)$ where $p = \mathbb{E}[s(\boldsymbol{x} + \boldsymbol{\epsilon}, y)]$. Originally, RSCP did not account for finite sample correction making its coverage guarantee only asymptomatically valid. Yan et al. (2024) show that correcting for finite samples in RSCP leads to trivial prediction sets $\bar{\mathcal{C}}(\boldsymbol{x}) = \mathcal{Y}$. As a remedy, they define a new score function based on temperature scaling which in combination with conformal training (Stutz et al., 2021) improves the average set size. So far both methods use test-time robustness. In § E, we show that BinCP outperforms RSCP+.

In contrast Zargarbashi et al. (2024) utilizes the CDF structure of the score and instead apply the tighter CDF-based bound defining CDF aware sets (CAS). In combination with calibration-time robustness, they show that only $|\mathcal{D}_{\text{cal}}|$ certificate bounds should be computed to maintain a robust coverage guarantee as in Eq. 1. In addition to a gain in computational efficiency, they show that in the calibration-time robustness, the error correction budget can be used more efficiently. On CIFAR-10 they return a relatively small conformal set size. In all aforementioned methods, a large MC sampling budget (e.g. $10^4$ samples) is assumed which is challenging for real-time applications. This issue is exacerbated for datasets like ImageNet where the large number of classes amplifies the effect of multiple testing corrections.

**Robust CP via verifiers.** Outside the scope of randomized smoothing Jeary et al. (2024) use neural network verification to compute upper (or lower) bounds. This requires white-box access to the model weights, while our proposed method works for any black-box model and randomized or exact smoothing-based certificate. Interestingly, in (Jeary et al., 2024) (Table 1) the empirical evaluation is for $r = 0.02$ which is smaller than the minimum radius we reported. For completeness, we evaluated BinCP on very small radii in § E (Fig. 9-left), and for the same $r$ our sets are $2\times$ smaller. As discussed in § 6 (Fig. 6) in general, smooth prediction, even without accounting for the adversary, shows to have an inherent robustness.

**Other robustness results.** Alternatively Ghosh et al. (2023) introduce probabilistic robust coverage which intuitively accounts for *average* adversarial inputs. This is in contrast with our core assumption of worst-case adversarial inputs. In other words, instead of $1 - \alpha$ coverage for any point within the perturbation ball around $\boldsymbol{x}_{n+1}$, "probabilistically robust coverage" guarantees that the probability to cover the true label remains above $1 - \alpha$ on average over all $\tilde{\boldsymbol{x}} \in \mathcal{B}(\boldsymbol{x})$, while we consider the worst-case $\tilde{\boldsymbol{x}}$. Their "quantile of quantiles" method looks superficially similar to BinCP as they also compute $n+k+1$ quantiles. However there are two notable differences. Their first order of quantiles (on true calibration scores and the score for each class of the test point) is over random draws from the perturbation set. BinCP computes the first order quantiles ($\tau_i(p)$ in fixed $\tau$ setup) over the smooth score distribution. Their conservative quantile index is based on a user-specified hyperparameter that accounts for conservativeness while BinCP finds the certified probability $\mathrm{c}^{\downarrow}[p_\alpha, \mathcal{B}]$ for the worst case adversarial example. BinCP guarantees that any $\tilde{\boldsymbol{x}} \in \mathcal{B}(\boldsymbol{x})$ is covered if $\boldsymbol{x}$ is covered. Furthermore, there are other works addressing distribution or covariate shift in general beyond the score of worst-case noise robustness (Barber et al., 2022; Tibshirani et al., 2019).

## 8 CONCLUSION

We introduce BinCP, a robust conformal prediction method based on randomized smoothing that produces small prediction sets with a few Monte-Carlo samples. The key insight is that we binarize the distribution of smooth scores, by a threshold (or thresholds) that maintains the coverage guarantee. We show that both calibration and test-time robustness approaches (discussed in § 2) are equivalent to computing a single binary certificate. This directly enables us to use any certificate that returns a certified lower-bound probability right out of the box; including de-randomized certificate for $\ell_1$ norm. The binarization enables us to use tighter Clopper-Pearson confidence intervals. This leads directly to faster computation of prediction sets with a significantly lower Monte Carlo sample-rate (compared to the SOTA), and therefore less forward passes per input. Interestingly, we show that even without accounting for an adversarial setup, CP with smooth score shows more robustness to adversarial examples in comparison with the conventional vanilla CP.

ACKNOWLEDGMENT

We thank Giuliana Thomanek and Jimin Cao for their feedbacks on our initial draft.

ETHICS STATEMENT

In this paper, we study the robustness of conformal prediction. The main focus of our work is to increase the reliability of conformal prediction in presence of noise or adversarial perturbations. Therefore, we don't see any particular ethical concern to mention about this study.

REPRODUCIBILITY STATEMENT

To ensure the reproducibility of our results, we have provided the algorithm is in § A, and the implementations are available at the BinCP Github repository. The models we used are also pre-trained and all accessible from the cited works. We specified the setup including parameter selections in § 6.

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

## A  ALGORITHM FOR ROBUST (AND VANILLA) BINCP

Here we provide the algorithm for BinCP in both $p$-fixed, and $\tau$-fixed setups. The two setups differ in calibration and finite sample correction, while computing the certificate, and returning prediction sets is similar in both. Note that in $p$-fixed version after computing the quantile $\tau_\alpha$ we correct for finite samples which results in a lower $p_\alpha^\downarrow$.

Note that in both algorithms we set $m$, and $\eta$ as fixed hyper-parameters defining the number of random samples per each datapoint and the collective failure probability of the confidence intervals. Therefore $\mathrm{ClopperPearson}(p)$ actually refers to the Clopper-Pearson interval with $p \cdot m$ success out of $m$ samples and failure probability of $\eta/(k + |\mathcal{D}_{\mathrm{cal}}|)$.

---

**Algorithm 1:** BinCP with $\tau$-fixed setup

---

**Input:** Score function $s : \mathcal{X} \times \mathcal{Y} \to \mathbb{R}$; Calibration set $\mathcal{D} = \{\boldsymbol{x}_i, y_i\}_{i=1}^n$; Smoothing scheme $\xi$;
       Threat model $\mathcal{B}$ satisfying the assumption in Lemma 2; Fixed threshold $\tau$, and
       (potentially perturbed) test point $\tilde{\boldsymbol{x}}_{n+1}$.
**Output:** A prediction set $\bar{\mathcal{C}}_{\mathrm{bin}}(\tilde{\boldsymbol{x}}_{n+1})$ with $1 - \alpha$ robust coverage probability
**for** *each calibration point* $(\boldsymbol{x}_i, y_i) \in \mathcal{D}_{\mathrm{cal}}$ **do**
     Sample from $\boldsymbol{x}_i + \boldsymbol{\epsilon}$ for $m$ times;
     Compute $q_i = \frac{1}{m} \sum_{j=1}^m \mathbb{I}[s(\boldsymbol{x}_i + \boldsymbol{\epsilon}, y_i) \geq \tau]$;
     **if** *Exact Certificate* **then**
         $q_i^\downarrow := q_i$;
     **else**
         $q_i^\downarrow := \mathrm{ClopperPearson}_{\mathrm{low}}(q_i)$;
     **end**
**end**
Set $p_\alpha^\downarrow = \mathbb{Q}\left(\alpha; \{q_i^\downarrow\}_{i=0}^n\right)$;
Compute $\mathrm{c}^\downarrow[p_\alpha^\downarrow, \mathcal{B}]$ from Eq. 5 (Lower bound minimization);
**for** *each class* $y \in \mathcal{Y}$ **do**
     Sample from $\xi(\tilde{\boldsymbol{x}}_{n+1})$ for $m$ times;
     Compute $q_{n+1,y} = \frac{1}{m} \sum_{j=1}^m \mathbb{I}[s(\tilde{\boldsymbol{x}}_{n+1} + \boldsymbol{\epsilon}, y) \geq \tau]$;
     **if** *Exact Certificate* **then**
         $q_{n+1,y}^\uparrow := q_{n+1}$;
     **else**
         $q_{n+1,y}^\uparrow := \mathrm{ClopperPearson}_{\mathrm{high}}(q_{n+1,y})$;
     **end**
**end**
**return** $\bar{\mathcal{C}}\{y : q_{n+1,y}^\uparrow \geq \mathrm{c}^\downarrow[p_\alpha^\downarrow, \mathcal{B}]\}$

---

**Score functions.** Throughout the paper we reported results for TPS score – directly setting the softmax values as the score $s(\boldsymbol{x}, y) = \pi(\boldsymbol{x}, y)$ (Sadinle et al., 2018). In vanilla CP ($r = 0$), TPS tends to over-cover easy examples and under-cover hard ones (Angelopoulos & Bates, 2021). Alternatively "adaptive prediction sets" (APS) aiming for conditional coverage uses the score function defined as $s(\boldsymbol{x}, y) := -\left(\rho(\boldsymbol{x}, y) + u \cdot \pi(\boldsymbol{x})_y\right)$ where $\rho(\boldsymbol{x}, y) := \sum_{c=1}^K \pi(\boldsymbol{x})_c 1\left[\pi(\boldsymbol{x})_c > \pi(\boldsymbol{x})_y\right]$ is the sum of all classes predicted as more likely than $y$, and $u \in [0, 1]$ is a uniform random value that breaks the ties between different scores to allow exact $1 - \alpha$ coverage (Romano et al., 2020). Another approach is to directly use the logits of the model as the score. This is not applicable in CAS and RCSP+ (Zargarbashi et al., 2024; Yan et al., 2024) as they work with bounded scores. BinCP can also work with unbounded score, hence, we also report the results on CP with logits. All three score functions are reported in Fig. 6-left for BinCP. Interestingly we do not see any significant difference in set size between APS and TPS when smoothed. We report BinCP with APS score in § E over all $\sigma$, and radii. Orthogonally, while RSCP with either scores quickly breaks to returning trivial sets, RSCP+ refines the score function through a biased temperature scaling. This can also be considered as another score function (or transformation over any score function) tailored for robust setup. We compare BinCP with RSCP+ in Fig. 14 (§ E).

---

**Algorithm 2:** BinCP with $p$-fixed setup

---

**Input:** Data, score function, smoothing, and $\mathcal{B}$ same as algorithm 1. Fixed probability $p_\alpha$
**Output:** Same as algorithm 1
Update $p_\alpha \leftarrow \lceil p_\alpha \cdot m \rceil / m$ (accounting for discrete samples);
**for** *each calibration point* $(\boldsymbol{x}_i, y_i) \in \mathcal{D}_{\mathrm{cal}}$ **do**
  Sample from $\boldsymbol{x}_i + \boldsymbol{\epsilon}$ for $m$ times;
  Compute $\tau_i = \mathbb{Q}\left(p_\alpha; \{s(\boldsymbol{x}_i + \boldsymbol{\epsilon}, y_i)\}_{j=1}^m\right)$;
  **if** *Exact Certificate* **then**
   $p_\alpha^\downarrow := p_\alpha$;
  **else**
   $p_\alpha^\downarrow := \mathrm{ClopperPearson}_{\mathrm{low}}(p_\alpha)$;
  **end**
**end**
Set $\tau_\alpha = \mathbb{Q}\left(\alpha; \{\tau_i\}_{i=0}^n\right)$;
Compute $\mathrm{c}^\downarrow[p_\alpha^\downarrow, \mathcal{B}]$ from Eq. 5 (Lower bound minimization);
**for** *each class* $y \in \mathcal{Y}$ **do**
  Sample from $\xi(\tilde{\boldsymbol{x}}_{n+1})$ for $m$ times;
  Compute $q_{n+1,y}^\uparrow = \frac{1}{m}\sum_{j=1}^m \mathbb{I}[s(\tilde{\boldsymbol{x}}_{n+1} + \boldsymbol{\epsilon}, y) \geq \tau_\alpha]$;
  $q_{n+1,y}^\uparrow :=$ same as algorithm 1;
**end**
**return** $\bar{\mathcal{C}}\{y : q_{n+1,y}^\uparrow \geq \mathrm{c}^\downarrow[p_\alpha^\downarrow, \mathcal{B}]\}$

---

# B COMPUTING CERTIFICATE OPTIMIZATION

**Canonical view.** Turns out that for isotropic Gaussian and sparse smoothing, we can always attain this minimum at canonical points – for any test point $\boldsymbol{x}_{n+1}$ we can translate the function, and the points to the origin, and the worst case to the border of $\mathcal{B}(\boldsymbol{x}_{n+1})$. Formally, there is a pair $(\boldsymbol{u}, \tilde{\boldsymbol{u}})$ such that $\rho_{\boldsymbol{u}, \tilde{\boldsymbol{u}}} = \rho_{\boldsymbol{x}_{n+1}, \tilde{\boldsymbol{x}}_{n+1}}$ for any $\boldsymbol{x}_{n+1}$, and $\tilde{\boldsymbol{x}}_{n+1} \in \mathcal{B}(\boldsymbol{x}_{n+1})$. Namely for the continuous ball $\mathcal{B}_r$ the canonical vectors are $\boldsymbol{u} = \boldsymbol{0}$ and $\tilde{\boldsymbol{u}} = [r, 0, 0, \dots]$. For the binary $\mathcal{B}_{r_a, r_d}$ we have the canonical $\boldsymbol{u} = [0, \dots, 0, 1, \dots, 1]$ and $\tilde{\boldsymbol{u}} = \boldsymbol{1} - \boldsymbol{u}$ where $\|\boldsymbol{u}\|_0 = r_d$ and $\|\tilde{\boldsymbol{u}}\|_0 = r_a$. Intuitively it is due to the symmetry of the ball and the smoothing distribution. To avoid many notations, we again use the $\boldsymbol{x}$, and $\tilde{\boldsymbol{x}}$ in the rest of the discussion that refers to the canonical points.

To obtain an upper or lower bound (Eq. 5 as maximization or minimization) we partition the space $\mathcal{X}$ to regions where the likelihood ratio between $(\boldsymbol{x}, \tilde{\boldsymbol{x}})$ is constant; formally $\mathcal{X} = \cup_{i=1}^k \mathcal{R}_i$ where $\forall \boldsymbol{z} \in \mathcal{R}_i : \Pr[\xi(\boldsymbol{x}) = \boldsymbol{z}] / \Pr[\xi(\tilde{\boldsymbol{x}}) = \boldsymbol{z}] = c_i$. For any function $h$ we can find an equivalent piecewise-constant $\hat{h}$ where inside each region it is assigned to the expected value of $h$ in that region. Let $t_i = \Pr[\xi(\boldsymbol{x}) = \boldsymbol{z}]$, and $\tilde{t}_i = \Pr[\xi(\tilde{\boldsymbol{x}}) = \boldsymbol{z}]$ then Eq. 5 simplifies to the following linear programming

$$\min_{\boldsymbol{h} \in [0,1]^k} \boldsymbol{h}^\top \tilde{\boldsymbol{t}} \quad \text{s.t.} \quad \boldsymbol{h}^\top \boldsymbol{t} = p_\alpha \tag{6}$$

Where $\boldsymbol{h}$, $\boldsymbol{t}$, and $\tilde{\boldsymbol{t}}$ are vectors that include the values $h_i$, $t_i$, $\tilde{t}_i$ for each region. The optimum solution to the simplified linear programming is obtained by sorting regions based on the likelihood ratio and greedily assigning $h$ to the possible maximum in each region until the budget $\boldsymbol{h}^\top \boldsymbol{t} = p_\alpha$ is met. The rest of the regions are similarly assigned to zero. This problem is equivalent to fractional knapsack. For isotropic Gaussian smoothing Cohen et al. (2019) show that the optimal solution has a closed form $\rho_\alpha = \Phi_\sigma(\Phi_\sigma^{-1}(p_\alpha) - r)$ where $\Phi_\sigma$ is the Gaussian CDF function of the Gaussian distribution with standard deviation $\sigma$. For sparse smoothing, following Bojchevski et al. (2020) we solve the greedy program on at most $r_a + r_d + 1$ distinct regions. The runtime is linear w.r.t. to the add and delete budget. For more detailed explanation see (Lee et al., 2019; Bojchevski et al., 2020).

# C  SUPPLEMENTARY TO THEORY

## C.1  VANILLA BINCP

**Conformal risk control.** We use conformal risk control (CRC) (Angelopoulos et al., 2022) to prove the coverage guarantee in BinCP. Here we succinctly recall it before the proof of Prop. 1.

**Theorem 2** (Conformal Risk Control - rephrased). *Let $\lambda$ be a parameter (larger $\lambda$ yields more conservative output), and $L_i : \Lambda \to (-\infty, B]$ for $i = 1, \ldots, n+1$ be exchangeable random functions. If (i) $L_i$s are non-increasing right-continuous w.r.t. $\lambda$, (ii) for $\lambda_{\max} = \sup \Lambda$ we have $L_i(\lambda_{\max}) \leq \alpha$, and (iii) $\sup_\lambda L_i \leq B < \infty$, then we have:*

$$\mathbb{E}[L_{n+1}(\hat{\lambda})] \leq \alpha \quad for \quad \hat{\lambda} = \inf \left\{ \lambda : \frac{\sum_{i=1}^n L_i(\lambda)}{n+1} + \frac{B}{n+1} \leq \alpha \right\} \tag{7}$$

In case that $B = 1$, by simplifying Eq. 7, we have $\hat{\lambda} = \inf \{\lambda : \sum_{i=1}^n L_i(\lambda) \leq \alpha(n+1) - 1\}$ We use this framework to prove the guarantee in BinCP.

*Proof to Prop. 1.* We prove the theorem through re-parameterizing of the conservativeness variable in each case. For fixed $p$ we set $\tau = 1 - \lambda$; similarly for fixed $\tau$ we set $p = -\lambda$. In both cases, the risk is defined as

$$L_i(\tau, p) = 1 - \text{accept}(\boldsymbol{x}_i, y_i; p, \tau)$$

which for simplicity we define $\text{reject}(\boldsymbol{x}_i, y_i; p, \tau) = 1 - \text{accept}(\boldsymbol{x}_i, y_i; p, \tau)$ and by definition we have $\text{reject}(\boldsymbol{x}_i, y_i; p, \tau) = \mathbb{I}[\Pr[s(\boldsymbol{x} + \boldsymbol{\epsilon}, y) < \tau] > 1 - p] = \mathbb{I}[\Pr[s(\boldsymbol{x} + \boldsymbol{\epsilon}, y) \geq \tau] < p]$. We show that the risk function satisfies the properties for a risk function feasible to the setup in Theorem 2.

1. **Non-increasing to $\lambda$.** In both cases the risk $L_i$ is non-inscreasing to $\lambda$; for fixed $p$ we have

$$\lambda_1 < \lambda_2 \Rightarrow 1 - \lambda_1 > 1 - \lambda_2$$
$$\Rightarrow \Pr[s(\boldsymbol{x} + \boldsymbol{\epsilon}, y) < 1 - \lambda_1] \geq \Pr[s(\boldsymbol{x} + \boldsymbol{\epsilon}, y) < 1 - \lambda_2]$$
$$\Rightarrow \text{reject}(\boldsymbol{x}, y; p, 1 - \lambda_1) \geq \text{reject}(\boldsymbol{x}, y; p, 1 - \lambda_2)$$

   Now for fixed $\tau$, let $p_{\boldsymbol{x}} = \Pr[s(\boldsymbol{x} + \boldsymbol{\epsilon}, y) \geq \tau]$ then we have

$$\lambda_1 < \lambda_2 \Rightarrow p_1 > p_2 \text{ means that } \mathbb{I}[p_{\boldsymbol{x}} \leq p_1] \geq \mathbb{I}[p_{\boldsymbol{x}} \leq p_2]$$
$$\Rightarrow \text{reject}(\boldsymbol{x}, y; -\lambda_1, \tau) \geq \text{reject}(\boldsymbol{x}, y; -\lambda_2, \tau)$$

   Intuitively by adapting the definition of the rejection (risk) function $\text{reject}(\boldsymbol{x}_i, y_i; p, \tau) = \mathbb{I}[\Pr[s(\boldsymbol{x} + \boldsymbol{\epsilon}, y) \geq \tau] < p]$, if we increase $\lambda$ which means decreasing $p$, the chance of rejecting a label decreases. This is because, we require the same probability mass to be lower than a smaller value.

2. **Right continuous.** Formally the function accept is

$$\text{accept}(\boldsymbol{x}, y; p, \tau) = \begin{cases} 1 & \text{if } \Pr[s(\boldsymbol{x} + \boldsymbol{\epsilon}, y) \geq \tau] \geq p \\ 0 & \text{otherwise} \end{cases}$$

   Across the domain (for either $p$ or $\tau$) this function has two values and it is just non-continuous in the jump between the values. For both $p$ and $\tau$ this function is left continuous due to the $\geq$ comparison. Therefore for fixed $p$ the function $\text{reject}(\boldsymbol{x}, y; p, 1 - \lambda)$ is right continuous to $\lambda = 1 - \tau$. Similar argument follows for fixed $\tau$.

3. **Feasibility of risks less than $\alpha$.** For fixed $p > 0$ if we set $\lambda = 1 - \tau$ to $\infty$ ($\tau = -\infty$), for all $\boldsymbol{x}_i$, we have $\text{accept}(\boldsymbol{x}_i, y_i; p, 0) = 1$; i.e. the risk is 0 for every data. Similarly by approaching $p$ to zero in fixed $\tau$ setup, we decrease the risk to 0 for everyone. To avoid corner cases we can restrict $\tau$ to $\max s(\boldsymbol{x} + \boldsymbol{\epsilon}, y)$ for $x \in \mathcal{X}$ from above.

4. **Limited upperbound risk.** For any parameter and any input the highest possible risk is in case of rejection which is 1 ($B = 1$).

**Fixed $p$.** The risk function $L_i(\lambda) = \text{reject}(\boldsymbol{x}_i, y_i; p, 1 - \lambda)$ which means that the prediction set $\mathcal{C}(\boldsymbol{x}_i; p, 1 - \lambda)$ excludes $y_i$. We have

$$\mathbb{E}[\text{reject}(\boldsymbol{x}_{n+1}, y_{n+1}; p, 1 - \hat{\lambda})] \leq \alpha \text{ for } \hat{\lambda} = \inf_\lambda \left\{ \lambda : \sum_{i=1}^n \text{reject}(\boldsymbol{x}_i, y_i; p, 1 - \lambda) \leq \alpha(n+1) - 1 \right\}$$

Setting back the $\tau = 1 - \lambda$, and rewriting the expectation as a probability form, we have

$$\Pr[y_{n+1} \in \mathcal{C}(\boldsymbol{x}_{n+1}; p, \tau_p)] \geq 1 - \alpha \text{ for } \tau_p = \sup_\tau \left\{ \tau : \sum_{i=1}^n \text{accept}(\boldsymbol{x}_i, y_i; p, \tau) \geq (1 - \alpha)(n+1) \right\}$$

In the above, we used the fact that if a test fails on $\alpha(n+1) - 1$ variables among the total of $n$ variables, it passes on $n - [\alpha(n+1) - 1]$ and $(1 - \alpha)(n+1) = n - [\alpha(n+1) - 1]$.

**Fixed $\tau$.** Similarly, we define the risk function as $L_i(\lambda) = \text{reject}(\boldsymbol{x}_i, y_i; -\lambda, \tau)$. We have

$$\mathbb{E}[\text{reject}(\boldsymbol{x}_{n+1}, y_{n+1}; p_\tau, \tau)] \leq \alpha \quad \text{for} \quad p_\tau = \inf_\lambda \left\{ \lambda : \sum_{i=1}^n \text{reject}(\boldsymbol{x}_i, y_i; -\lambda, \tau) \leq \alpha(n+1) - 1 \right\}$$

$\square$

*Proof to Lemma 1.* The function $\text{accept}(\boldsymbol{x}, y; p, \tau)$ is non-increasing in both $p$ and $\tau$. Therefore the term $\sum_{i=1}^n \text{accept}(\boldsymbol{x}_i, y_i; p, \tau)$ is also non-increasing in $p$ and $\tau$ and its range is the integer numbers between $0$ and $n$ (or $[n]$). For a fixed $p$, let $\tau_\alpha(p)$ be the solution to Eq. 2, then by definition it satisfies that

$$\sum_{i=1}^n \text{accept}(\boldsymbol{x}_i, y_i; p, \tau_\alpha(p)) \geq (1 - \alpha)(n+1)$$

This implies that $p$ satisfies the same condition for $p_\alpha(\tau_\alpha(p))$. Therefore $p_\alpha(\tau_\alpha(p)) \geq p$ as $p$ is a feasible solution in Eq. 3. The supremum search for $\tau_\alpha(p)$ directly implies that for any positive $\delta$ we have

$$\sum_{i=1}^n \text{accept}(\boldsymbol{x}_i, y_i; p, \tau_\alpha(p)) \geq \sum_{i=1}^n \text{accept}(\boldsymbol{x}_i, y_i; p, \tau_\alpha(p) + \delta) - 1$$

which intuitively means that increasing the $\tau(p)$ by any small margin fails at least in one more accept for calibration points. Since $\sum_{i=1}^n \text{accept}(\boldsymbol{x}_i, y_i; p, \tau_\alpha(p))$ is the sum of $n$ non-increasing functions, there is one index $i$ for which

$$\text{accept}(\boldsymbol{x}_i, y_i; p, \tau_\alpha(p)) = 1 \quad \text{and} \quad \text{accept}(\boldsymbol{x}_i, y_i; p, \tau_\alpha(p) + \delta) = 0$$

For any small positive $\delta$. Using the definition of the accept function we have

$$\Pr[s(\boldsymbol{x}_i + \boldsymbol{\epsilon}, y_i) \geq \tau_\alpha(p)] \geq p \quad \text{and} \quad \Pr[s(\boldsymbol{x}_i + \boldsymbol{\epsilon}, y_i) \geq \tau_\alpha(p) + \delta] < p$$

Due to the continuous strictly increasing CDF for $S_i$ we have $\Pr[s(\boldsymbol{x}_i + \boldsymbol{\epsilon}, y_i) \geq \tau(p)] = p$. Therefore for any small positive $\delta$

$$\text{accept}(\boldsymbol{x}_i, y_i; p, \tau_\alpha(p)) = 1 \quad \text{and} \quad \text{accept}(\boldsymbol{x}_i, y_i; p + \delta, \tau_\alpha(p)) = 0$$

which means that the accept function for $\boldsymbol{x}_i$ fails by adding a small number to $p$. Since all other accept functions are also non-increasing we have $\sum_{i=1}^n \text{accept}(\boldsymbol{x}_i, y_i; p, \tau_\alpha(p)) \leq (1 - \alpha)(n+1) - 1$. This implies that $p$ is also the supremum for $Eq.$ 2 with parameter $p_\alpha(\tau)$. $\square$

## C.2 ROBUST BINCP

*Proof to Lemma 2.* With $f_{\text{true}}(\boldsymbol{x}_i) = \mathbb{I}[s(\boldsymbol{x}_i, y_i) \geq \tau_\alpha]$ for true $y_i$, the calibration-time robust prediction set is defined as $\bar{\mathcal{C}}_{\text{cal}}(\tilde{\boldsymbol{x}}_{n+1}) = \{p(\tilde{\boldsymbol{x}}_{n+1}, y; \tau_\alpha) \geq \mathbb{Q}\left(\alpha; \{c^\downarrow[\bar{f}_{\text{true}}(\boldsymbol{x}_i), \mathcal{B}]\}_{i=1}^n\right)\}$. By definition we have $p_\alpha = \mathbb{Q}\left(\alpha; \{\bar{f}_{\text{true}}(\boldsymbol{x})\}_{i=1}^n\right)$. Both lower bound and upper bound functions are non-decreasing. As a result, the ranks, and hence the quantile index in $\{\bar{f}_{\text{true}}(\boldsymbol{x}_i)\}_{i=1}^n$ and $\{c^\downarrow[\bar{f}_{\text{true}}(\boldsymbol{x}_i), \mathcal{B}]\}_{i=1}^n$ are the same. Therefore, $\mathbb{Q}\left(\alpha; \{c^\downarrow[\bar{f}_{\text{true}}(\boldsymbol{x}_i), \mathcal{B}]\}_{i=1}^n\right) = c^\downarrow[p_\alpha, \mathcal{B}]$.

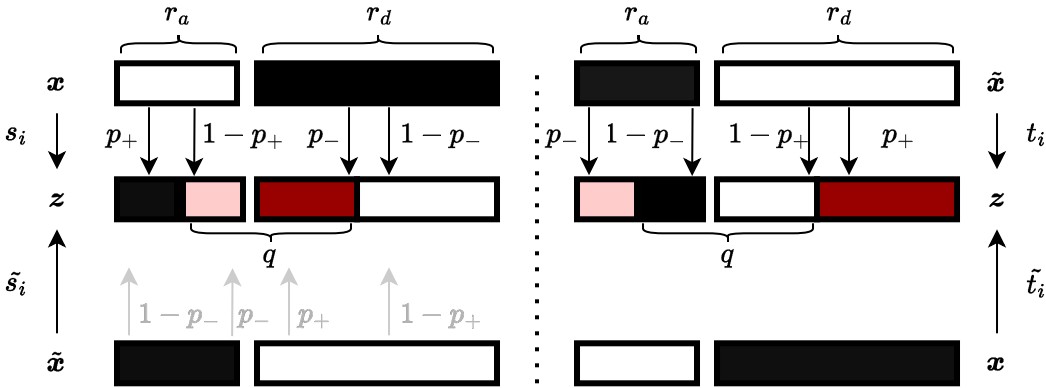

Figure 8: Illustration of likelihood ratio in sparse smoothing for both $\mathcal{B}_{r_a, r_d}$, and $\mathcal{B}_{r_d, r_a}$

The test-time robust prediction set is defined as $\bar{C}_{\text{test}} = \{y : \text{c}^{\uparrow}[\bar{f}_y(\tilde{\boldsymbol{x}}_{n+1}), \mathcal{B}^{-1}] \geq p_\alpha\}$, let $\tilde{p}_y = \bar{f}_y(\tilde{\boldsymbol{x}}_{n+1})$ then it follows

$$\text{c}^{\uparrow}[\bar{f}_y(\tilde{\boldsymbol{x}}_{n+1}), \mathcal{B}^{-1}] \geq p_\alpha \Leftrightarrow \text{c}^{\downarrow}[\text{c}^{\uparrow}[\bar{f}_y(\tilde{\boldsymbol{x}}_{n+1}), \mathcal{B}^{-1}], \mathcal{B}] \geq \text{c}^{\downarrow}[p_\alpha, \mathcal{B}]$$
$$\Leftrightarrow \bar{f}_y(\tilde{\boldsymbol{x}}_{n+1}) \geq \text{c}^{\downarrow}[p_\alpha, \mathcal{B}]$$

By definition $\text{accept}(\tilde{\boldsymbol{x}}_{n+1}, y; \text{c}^{\downarrow}[p_\alpha, \mathcal{B}], \tau_\alpha) = \mathbb{I}[\bar{f}_y(\tilde{\boldsymbol{x}}_{n+1}) \geq \text{c}^{\downarrow}[p_\alpha, \mathcal{B}]]$. $\qquad\square$

In the above, we proved that BinCP results in a valid conformal prediction. Here we prove the validity of robust BinCP to adversarial data within the bounded threat model.

*Proof to Lemma 3.* For each of the mentioned smoothing schemes we have:

**Gaussian smoothing.** In both cases since $p$ norm is symmetric for any point $\tilde{\boldsymbol{x}}$ it holds that $\|\tilde{\boldsymbol{x}} - \boldsymbol{x}\|_p \leq r$. In other words, from any perturbed point the clean point is within $\mathcal{B}_r(\tilde{\boldsymbol{x}})$. Therefore $\mathcal{B}_r^{-1} = \mathcal{B}_r$.

For simpler notation let $\bar{p} = \text{c}^{\uparrow}[p, \mathcal{B}_r]$. Given the closed from solution $\text{c}^{\downarrow}[p, \mathcal{B}_r] = \Phi_\sigma(\Phi_\sigma^{-1}(p) - r)$, and $\text{c}^{\uparrow}[p, \mathcal{B}_r] = \Phi_\sigma(\Phi_\sigma^{-1}(p) + r)$ we have

$$\bar{p} = \Phi_\sigma(\Phi_\sigma^{-1}(p) + r) \Leftrightarrow \Phi_\sigma^{-1}(\bar{p}) = \Phi_\sigma^{-1}(p) + r \Leftrightarrow \Phi_\sigma^{-1}(\bar{p}) - r = \Phi_\sigma^{-1}(p)$$
$$\Leftrightarrow \Phi_\sigma(\Phi_\sigma^{-1}(\bar{p}) - r) = p \Leftrightarrow \text{c}^{\downarrow}[\bar{p}, \mathcal{B}] = p$$

**Uniform smoothing.** For the uniform smoothing from Levine & Feizi (2021) we have that the smooth classifier is $1/(2\lambda)$-Lipschitz continuous. From that the certified lower and upper bounds are defined as $\text{c}^{\downarrow}[p, \mathcal{B}_r] = p - r \cdot 1/(2\lambda)$, and $\text{c}^{\uparrow}[p, \mathcal{B}_r] = p + r \cdot 1/(2\lambda)$ Therefore

$$\text{c}^{\downarrow}[\text{c}^{\uparrow}[p, \mathcal{B}_r^{-1}], \mathcal{B}_r] = \text{c}^{\downarrow}[p + \frac{r}{2\lambda}, \mathcal{B}_r] = p + \frac{r}{2\lambda} - \frac{r}{2\lambda} = p$$

A similar argument can be applied to any certificate that directly adds (or subtracts) the Lipschitz constant of the smooth classifier to the base probability.

**Sparse smoothing.** Any $\tilde{\boldsymbol{x}} \in \mathcal{B}_{r_a, r_d}(\boldsymbol{x})$ has at most $r_a$ zero bits, and $r_d$ one bits toggled from $\boldsymbol{x}$. By toggling those bit back we can reconstruct $\boldsymbol{x}$. The maximum needed toggles is therefore $r_d$ zero bits and $r_a$ one bits which is the definition of $\mathcal{B}_{r_d, r_a}$.

As discussed in § B, canonical points for $\mathcal{B}_{r_a, r_d}$ are $\boldsymbol{x} = [0, \dots, 0, 1, \dots, 1]$ and $\tilde{\boldsymbol{x}} = \mathbf{1} - \boldsymbol{x}$ where $\|\boldsymbol{x}\|_0 = r_d$ and $\|\tilde{\boldsymbol{x}}\|_0 = r_a$. For $\mathcal{B}_{r_a, r_d}^{-1}$ the canonical points are $\boldsymbol{u}, \tilde{\boldsymbol{u}}$ where $\|\boldsymbol{u}\|_0 = r_a$. By applying a permutation over $\boldsymbol{u}, \tilde{\boldsymbol{x}}$ and every other point in all regions we can set $\boldsymbol{u} = \tilde{\boldsymbol{x}}$, and $\tilde{\boldsymbol{u}} = \boldsymbol{x}$. For computing both $\mathcal{B}_{r_a, r_d}$, and $\mathcal{B}_{r_a, r_d}^{-1}$ there are $r_a + r_d + 1$ regions of constant likelihood ratio, each including all points that have the same number of total flips from the source $\boldsymbol{x}$, or $\boldsymbol{u}$;

formally $\mathcal{R}_q = \{ \boldsymbol{z} : \| \boldsymbol{x} - \boldsymbol{z} \|_0 = q \}$. The same region can also defined to preserve $r_d + r_a - q$ bits from $\tilde{\boldsymbol{x}}$. With $\frac{s_q}{\tilde{s}_q}$ as the likelihood ratio of a point $\boldsymbol{z}$ in $\mathcal{B}_{r_a, r_d}$ and $q = q_a + q_d$ as the number of changes in 1 and 0 bit, we have $s_q = (p_+)^{q_a}(1 - p_+)^{r_a - q_a}(p_-)^{q_d}(1 - p_-)^{r_d - q_d}$, and similarly $\tilde{s}_q = (p_-)^{q_a}(1 - p_-)^{r_a - q_a}(p_-)^{q_d}(1 - p_-)^{r_d - q_d}$. Then the likelihood ratio is simplified to

$$\frac{s_q}{\tilde{s}_q} = \left[ \frac{p_+}{1 - p_-} \right]^{q - r_d} \left[ \frac{p_-}{1 - p_+} \right]^{q - r_a} \tag{8}$$

As illustrated in Fig. 8 regions for $\mathcal{B}_{r_a, r_d}^{-1}$ are same as $\mathcal{B}_{r_a, r_d}$ only with reverse order. In other word, let $t_i, \tilde{t}_i$ be the probability of visiting region $\mathcal{R}_q$ from $\boldsymbol{u}$ and $\tilde{\boldsymbol{u}}$, then $t_i = \tilde{s}_{r_a + r_d + 1 - q}$, and $\tilde{t}_i = s_{r_a + r_d + 1 - q}$. For a fixed $\boldsymbol{z}$ the probability to visit $\boldsymbol{z}$ from $\boldsymbol{x}$ is the probability of toggling $q = \| \boldsymbol{z} - \boldsymbol{x} \|_0$ bits which is the same as toggling $q$ bits from $\tilde{\boldsymbol{u}}$ as $\tilde{\boldsymbol{u}} = \boldsymbol{x}$.

Solutions to $\mathrm{c}^{\downarrow}[p, \mathcal{B}_{r_a, r_d}]$ and $\mathrm{c}^{\uparrow}[p, \mathcal{B}_{r_a, r_d}^{-1}]$ are obtained from the following optimization functions:

$$\mathrm{c}^{\downarrow}[p, \mathcal{B}_{r_a, r_d}] = \min_{\boldsymbol{h} \in [0,1]^{r_a + r_d + 1}} \boldsymbol{h}^{\top} \tilde{\boldsymbol{t}} \quad \text{s.t.} \quad \boldsymbol{h}^{\top} \boldsymbol{t} = p$$

$$\mathrm{c}^{\uparrow}[p, \mathcal{B}_{r_a, r_d}^{-1}] = \max_{\boldsymbol{h} \in [0,1]^{r_a + r_d + 1}} \boldsymbol{h}^{\top} \tilde{\boldsymbol{s}} \quad \text{s.t.} \quad \boldsymbol{h}^{\top} \boldsymbol{s} = p$$

The solution to the lower bound optimization is obtained by a greedy algorithm. We visit each in increasing order w.r.t. $\frac{s_q}{\tilde{s}_q}$, we assign $h_q = 1$ until the budget $\boldsymbol{h}^{\top} \boldsymbol{s}$ is met and we set $h_q = 0$ for the remaining regions (fractional knapsack problem). For the maximization we do the same but in a decreasing order.

We want to prove $\mathrm{c}^{\downarrow}[\mathrm{c}^{\uparrow}[p, \mathcal{B}_{r_a, r_d}^{-1}], \mathcal{B}_{r_a, r_d}] = p$. This is the solution to

$$\min_{\boldsymbol{h} \in [0,1]^{r_a + r_d + 1}} \boldsymbol{h}^{\top} \tilde{\boldsymbol{t}} \quad \text{s.t.} \quad \boldsymbol{h}^{\top} \boldsymbol{t} = \mathrm{c}^{\uparrow}[p, \mathcal{B}_{r_a, r_d}^{-1}] = \boldsymbol{h}'^{\top} \tilde{\boldsymbol{s}}$$

Let $\overleftarrow{\tilde{\boldsymbol{s}}}$ be the vector $\tilde{\boldsymbol{s}}$ in reverse order. Then $\boldsymbol{t} = \overleftarrow{\tilde{\boldsymbol{s}}}$. From the problem definition we have that $\boldsymbol{h}'$ (the solution from maximization problem in reverse order) is a feasible solution. Given that the solution of the optimization (reduced to fractional knapsack problem) is always in form of $[1, \ldots, 1, \delta, 0, \ldots, 0]$ for some $\delta \in [0, 1]$, any vector of this form that satisfies the constraint is optimal. Therefore $\overleftarrow{\boldsymbol{h}'}$ is the solution to the maximization greedy problem. So the optimal solution is $\overleftarrow{\boldsymbol{h}'}^{\top} \tilde{\boldsymbol{t}} = \overleftarrow{\boldsymbol{h}'}^{\top} \overleftarrow{\tilde{\boldsymbol{s}}} = p$. □

For any $\ell_p$ with the same argument as $\ell_2$ ball we have $\mathcal{B}_r^{-1} = \mathcal{B}_r$. Similar to isotropic Gaussian smoothing, the Lipschitz continuity in DSSN-smoothed distribution shows that Lemma 3 applies to $\ell_1$ ball and this distribution as well.

## C.3 CORRECTION FOR FINITE SAMPLE MONTE-CARLO ESTIMATION

*Proof to Prop. 2.* With $p_i = \Pr[s(\boldsymbol{x}_i + \boldsymbol{\epsilon}, y_i) \geq \tau_{\alpha}]$ as the true probability of crossing $\tau_{\alpha}$ for each true score distribution in calibration set. We have $p_{\alpha} = \mathbb{Q}(\alpha; \{p_i\}_{i=1}^n)$. For all $i$ we have $q_i^{\downarrow} \leq p_i$ from which follows $p_{\alpha}^{\downarrow} \leq p_{\alpha}$. The probability of failure in each calibration datapoint is $\eta/(|\mathcal{D}_{\mathrm{cal}}| + k)$; as a result, from the union bound the probability of failure $q_i^{\downarrow} \leq p_i$ for all $i \in \{1, ..., n\}$ and therefore the quantile is $|\mathcal{D}_{\mathrm{cal}}|\eta/(|\mathcal{D}_{\mathrm{cal}}| + k)$.

For all classes of the test point we have $\tilde{q}_{n+1,y}^{\uparrow} \geq \tilde{p}_{n+1,y}$ with $\eta/(|\mathcal{D}_{\mathrm{cal}}| + k)$. Therefore, for the true class we have $\tilde{q}_{n+1} \geq \tilde{p}_{n+1}$ with $k\eta/(|\mathcal{D}_{\mathrm{cal}}| + k)$.

Conformal guarantee implies that with $1 - \alpha + \eta$ probability we have $p_{n+1} \geq p_{\alpha}$. The robustness certificate implies that $\tilde{p}_{n+1} \geq \mathrm{c}^{\downarrow}[p_{\alpha}, \mathcal{B}]$. Following holds by using the mentioned inequality:

$$\tilde{q}_{n+1}^{\uparrow} \underset{1 - \frac{k\eta}{n+k}}{\geq} \tilde{p}_{n+1} \underset{1 - \alpha + \eta}{\geq} \mathrm{c}^{\downarrow}[p_{\alpha}, \mathcal{B}] \underset{1 - \frac{n\eta}{n+k}}{\geq} \mathrm{c}^{\downarrow}[p_{\alpha}^{\downarrow}, \mathcal{B}]$$

From the union bound it follows that the total failure probability is less than $\alpha$. □

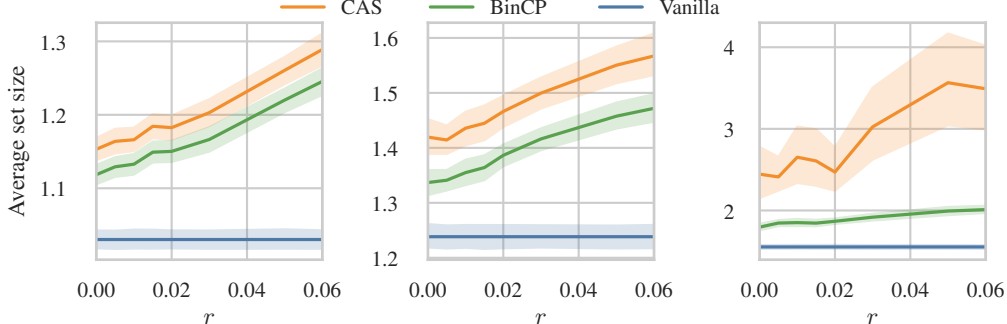

Figure 9: Comparison between BinCP and CAS on CIFAR-10 dataset with $\sigma = 0.25$ and small values of $r$. The nominal coverage $1 - \alpha$ is set to [From left to right] 85%, 90%, and 95%.

We estimate the probabilities in algorithm 1 following Prop. 2 directly. There, we estimate the $p_i = \Pr_{\boldsymbol{\epsilon}}[s(\boldsymbol{x}_i + \boldsymbol{\epsilon}, y_i) \geq \tau]$ via Monte Carlo samples resulting in $q_i = \frac{1}{m}\mathbb{I}[s(\boldsymbol{x}_i + \boldsymbol{\epsilon}, y_i) \geq \tau]$. Via Clopper Pearson bounds we find $q_i^{\downarrow}$ for which we have $q_i^{\downarrow} \leq p_i$ with adjusted $1 - \eta$ probability.

For the $p$-fixed approach (algorithm 2), we compute the discrete quantile $\tau_i = \mathbb{Q}(\lceil p_\alpha \cdot m \rceil / m; \{s(\boldsymbol{x}_i + \boldsymbol{\epsilon}, y_i)\}_{i=1}^m$ over the randomly sampled scores. By definition of discrete quantile function, without counting again, we have $\frac{1}{m}\mathbb{I}[s(\boldsymbol{x}_i + \boldsymbol{\epsilon}, y_i) \geq \tau_i] \geq p_\alpha$ ($p_\alpha$ proportion of the binary variables $\mathbb{I}[s(\boldsymbol{x}_i + \boldsymbol{\epsilon}, y_i) \geq \tau_i]$ are 1). Now, we bound a Bernoulli parameter that is the proportion of the distribution exceeding $\tau_i$. Again we use Clopper Pearson bound but this time on $p_\alpha$ which is known. Therefore after computing $\tau_\alpha$ we use $p_\alpha^{\downarrow}$ instead which is a minimum support for all $\tau_i$-s set as the $p_\alpha$ quantile of the sampled scores (with adjusted $1 - \eta$ confidence).

# D  Supplementary Discussion

## D.1  High-level understanding of robustness certificates

A certificate of robustness is a formal guarantee that the model predicts the same class for any perturbation within the specified threat model - within the ball around the input. In other words, if the function $f$ is certified to be robust for the point $\boldsymbol{x}$ w.r.t. $\mathcal{B}$, for any $\tilde{\boldsymbol{x}} \in \mathcal{B}(\boldsymbol{x})$ we have $\arg\max_y f_y(\boldsymbol{x}) = \arg\max_y f_y(\tilde{\boldsymbol{x}})$. This certificate ensures that the top label remains the same within the threat model (binary certificate). A similar (confidence, or soft) certificate guarantees a lower (and upper) bound on the model confidence within the threat model given the predictive probability. One way to attain such certificates is through verifiers. Verifiers need white-box access (knowledge about the model structure and weights) and they work efficiently only on a limited class of models. However, our robust conformal guarantee is black-box.

A common approach for black-box certification is through randomized smoothing. A randomly smoothed classifier results from inference given the input augmented with random noise. For example $g(\boldsymbol{x}) = \mathbb{E}_{\boldsymbol{\epsilon} \sim \mathcal{N}(\boldsymbol{0}, \sigma^2 \boldsymbol{I})}[f(\boldsymbol{x} + \boldsymbol{\epsilon})]$ – model $g$ returns the expected output of $f$ given randomly augmented $\boldsymbol{x}$ where the noise comes from an isotropic Gaussian distribution with scale $\sigma$. The randomization function is smooth even if the original function changes rapidly, which is the effect of the expectation. It is also Lipschitz continuous, meaning that we can bound the output based on the distance of $\tilde{\boldsymbol{x}}$ from $\boldsymbol{x}$. The latter allows us to provide formal guarantees that the top class probability (or confidence) remains high (changes slowly) even if $\tilde{\boldsymbol{x}} \in \mathcal{B}$ is passed to the model instead of $\boldsymbol{x}$.

Ultimately a randomized smoothing-based certificate returns a lower (or upper) bound probability (or score) on the expected output (given the randomized $\boldsymbol{x}$). In robust CP we use these bounds to answer "if instead of the clean input $\boldsymbol{x}_{n+1}$ which is already exchangeable with the calibration set, the model received the worst case $\tilde{\boldsymbol{x}}_{n+1} \in \mathcal{B}(\boldsymbol{x}_{n+1})$ how much lower the conformity score has become". Or in other words "if the model is queried with $\tilde{\boldsymbol{x}}_{n+1} \in \mathcal{B}(\boldsymbol{x}_{n+1})$ (which has a lower conformity score in order not to be covered) how much higher the conformity score of the clean

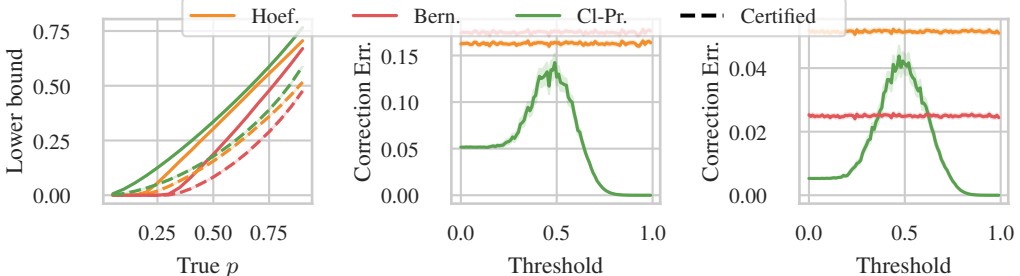

Figure 10: [Left] Confidence lower bound and the corresponding certified lower bound for scores derived from Beta and Bernoulli distributions. [Middle and right] Correction error (lower bound subtracted from the theoretical mean) of the scores distributed from the Gaussian distribution both in continuous case (mean lower bound) and binarized case (lower bound on the Bernoulli parameter) for [Middle] 100 and [Right] 1000 samples. Details of the experiments are in § D.2.

input can be". Technical details of smoothing-based certificates are mentioned in § 4. For a more detailed discussion see (Cohen et al., 2019; Kumar et al., 2020).

## D.2 COMPARISON OF CONFIDENCE INTERVALS

As discussed in § 5, BinCP, CAS, and RSCP, all require true probability, CDF, and mean from the distribution of scores which is intractable to compute (except in the case of de-randomized BinCP). Therefore we use confidence bounds that are lower (or higher) than the true values with collective probability $1 - \eta$ (which is taken into account while calibrating). CAS, and RSCP are defined on continuous scores that are bounded by Hoeffding, Bernstein, or DKW inequalities. BinCP is defined through binarized scores, and the final parameter is the success probability of a Bernoulli distribution which can bounded by the Clopper-Pearson interval which is exact (Clopper & Pearson, 1934). The width of all mentioned confidence intervals is decreasing w.r.t. the sample size. Therefore a tighter interval can result in the same or better efficiency (correction error) with fewer samples; e.g. For scores sampled from a Gaussian $\mathcal{N}(0.5, 0.1)$ Clopper Pearson error (for $z \geq 0.6$) with 100 samples is still lower than Bernstein's error with 250 samples.

To illustrate this we conducted two experiments. First, to compare the tightness of each concentration inequality, we sampled from a Beta distribution with mean $p$ to have continuous score values between $[0, 1]$. The distribution for a fixed $\beta$ is $\text{Beta}(\frac{p}{\beta(1-p)}, \beta)$. Then for the continuous score, we computed both Hoeffding's and Bernstein's lower bound on the mean, alongside the Clopper-Pearson bound for the given parameter $p$ and the same sample size. As shown in Fig. 10-left (with $\beta = 1$) the binary lower bound is always higher (better). Since the certified lower bound is an increasing function of the given probability, the certified lower bound for the binary values is again higher.

In another experiment shown in Fig. 10-middle and right we sample scores from a Gaussian distribution $\mathcal{N}(0.5, 0.1)$, and computed the lower-bound mean given both Hoeffding and Bernstein's inequalities. Then for various thresholds, we computed the probability of scores passing that threshold and lower bounded this probability by Clopper Pearson concentration inequality. As shown in the figure for lower sample rates, binarization results in less error compared to the theoretical mean. Even with higher sample rates Clopper Pearson interval is significantly tighter than the other two for low and high thresholds (which is a parameter of BinCP).

**Proposition 3.** *Let $X \sim \text{Beta}(a, b)$ and $x_1, \ldots, x_m$ be $m$ i.i.d. samples of $X$. Given the empirical mean $\bar{x} = \frac{1}{m} \sum_{i=1}^{m} x_i$ the upper bound for the true mean $\mu = \mathbb{E}[X]$ is given by the Hoeffding's inequality as $\mu \leq \bar{x} + b_{\text{hoef}}$, where $b_{\text{hoef}} = \sqrt{\frac{\ln\left(\frac{1}{\eta}\right)}{2m}}$. For any user-specified $\tau \in (0, 1)$, let $Y =*

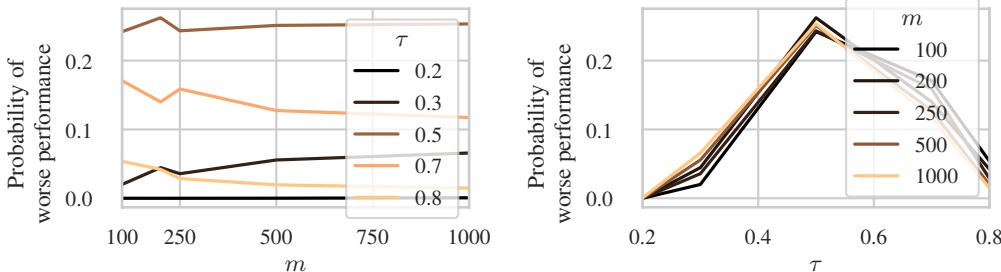

Figure 11: Probability of observing higher upper bound from Clopper Pearson confidence interval in comparison with Hoeffding's interval. The result is for $\text{Beta}(2, 2)$, and $\eta = 0.01$.

$\mathbb{I}[X > \tau]$. *The Clopper-Pearson upper bound $p_u$ for the true $p = \mathbb{E}[Y] = \Pr[X \geq \tau]$ is:*

$$p_u = \Phi_{\text{Beta}}^{-1}(1 - \eta; 1 + \sum_{i=1}^{m} \mathbb{I}[x_m > \tau], m - \sum_{i=1}^{m} \mathbb{I}[x_m > \tau_\mu])$$

*Each upper bound holds with probability $1 - \eta$. For any number of samples $m$, and any significance level $\eta$, the probability that the CP bound is tighther is $\Pr[p_u - \mathbb{E}[Y] \leq b_{\text{hoef}}]$ and equals:*

$$\Pr[p_u - \mathbb{E}[Y] \leq b_{\text{hoef}}] = \Phi_{\text{Binom}}(\hat{m}; m, \mathbb{E}[Y]) \tag{9}$$

*where $\hat{m}$ is defined in Eq. 11.*

*Proof.* The variable $Y$ is distributed as $Y \sim \text{Bernoulli}(p)$ where $p = \mathbb{E}[Y] = 1 - \Phi_{\text{Beta}}(\tau; a, b)$ and $\Phi_{\text{Beta}}$ is the CDF of the beta distribution with parameters $a$ and $b$.

Let $m_+ = \sum_{i=1}^{m} \mathbb{I}[x_m > \tau]$. We will compute the probability that the inequality

$$p_u - \mathbb{E}[Y] \leq \sqrt{\frac{\ln\left(\frac{1}{\eta}\right)}{2m}} \tag{10}$$

holds. Substituting the definition of $p_u$ and $\mathbb{E}[Y]$ we get:

$$\Phi_{\text{Beta}}^{-1}(1 - \eta; 1 + m_+, m - m_+) - (1 - \Phi_{\text{Beta}}(\tau; a, b)) \leq \sqrt{\frac{\ln\left(\frac{1}{\eta}\right)}{2m}} \Leftrightarrow$$

$$1 - \eta \leq \Phi_{\text{Beta}}\left(\sqrt{\frac{\ln\left(\frac{1}{\eta}\right)}{2m}} + 1 - \Phi_{\text{Beta}}(\tau; a, b); 1 + m_+, m - m_+\right) \Leftrightarrow$$

$$\eta \geq 1 - \Phi_{\text{Beta}}\left(\sqrt{\frac{\ln\left(\frac{1}{\eta}\right)}{2m}} + 1 - \Phi_{\text{Beta}}(\tau; a, b); 1 + m_+, m - m_+\right)$$

Define $\hat{m}$ as the break-point after which the Clopper-Pearson bound becomes looser than the Hoeffding bound:

$$\hat{m} = \sup\left\{m_+ : \mathbb{I}\left[\eta \geq 1 - \Phi_{\text{Beta}}\left(\sqrt{\frac{\ln\left(\frac{1}{\eta}\right)}{2m}} + 1 - \Phi_{\text{Beta}}(\tau; a, b); 1 + m_+, m - m_+\right)\right]\right\} \tag{11}$$

In other words, $m_+ > \hat{m} \Leftrightarrow p_u - \mathbb{E}[E] > b_{\text{hoef}}$. Since $\Phi$ is monotonic, it follows that:

$$\Pr[p_u - \mathbb{E}[Y] > b_{\text{hoef}}] = \Pr[m_+ > \hat{m}] = 1 - \Phi_{\text{Binom}}(\hat{m}; m, \mathbb{E}[Y]) \tag{12}$$

Where $\Phi_{\text{Binom}}$ is the CDF of the Binomial distribution with the specified parameters.

$\square$

Similarly, we can compare Clopper-Pearson bound with the Bernstein bound we use $\mu \leq \bar{x} + b_{\text{bern}}$ where

$$b_{\text{bern}} = \sqrt{2\sigma_m^2 \frac{\ln\left(\frac{2}{\eta}\right)}{m} + \frac{7\ln\left(\frac{2}{\eta}\right)}{3(m-1)}}$$

By replacing $b_{\text{hoef}}$ with $b_{\text{bern}}$ in Prop. 3 we can derive a similar result. We choose a Beta distribution to simulate the fact that conformity scores such as TPS and APS are bounded. Moreover, we need bounded scores to be able to apply Hoeffding's inequality. Any other distribution (after some transformation that ensures bounded scores) could be used, as long as we can compute its CDF.

In Fig. 11 we show the probability of Hoeffding bound being tighter than Clopper-Pearson bound (it's complement is defined in Eq. 12) for $X \sim \text{Beta}(2, 2)$ for different values of $m$ and $\tau$. There is a choice of $\tau$ such that the probability is effectively $0$ for all values of $m$, i.e. the CP bound is always better. Interestingly, at worst, both in terms of the number of samples $m$ and $\tau$, we see that it is less than 25%. That is, Clopper-Pearson bound is better on average for all configurations.

To get some additional intuition, instead of the exact Clopper-Pearson bound for $p$ we can use the following bound derived from a Normal approximation which approximately holds with probability $1 - \eta$:

$$p \leq \hat{p} + \frac{z_\eta}{\sqrt{m}}\sqrt{\hat{p}(1-\hat{p})}$$

where $\hat{p} = \frac{1}{m}\sum_{i=1}^{m}\mathbb{I}[x_m > \tau]$ and $z_\eta$ is the $1 - \eta$ quantile of the standard normal distribution. It is not difficult to verify that for all values of $\hat{p} \in [0, 1]$ we have that

$$\frac{z_\eta}{\sqrt{m}}\sqrt{\hat{p}(1-\hat{p})} \leq \sqrt{\frac{\ln\left(\frac{1}{\eta}\right)}{2m}}$$

To see this, note that the $\sqrt{m}$ term cancels, and for $\eta = 0.05$ $z_\eta \approx 1.64$, $\sqrt{\frac{\ln\left(\frac{1}{\eta}\right)}{2}} \approx 1.22$. Since $\sqrt{\hat{p}(1-\hat{p})} \in [0, 0.5]$, even in the worst-case $1.64 \cdot 0.5 \leq 1.22$. This analysis again confirms that CP gives tighter bounds. Prop. 3 can be analogues extended to lower bounds.

**Finite sample correction for fixed $\tau$ setup.** What we showed in Prop. 2 adds MC sample correction to BinCP with fixed $\tau$ computation. We can correct for finite samples in a fixed $p$ setup in a similar way. First, we compute the $\tau_i(p)$ for each of the calibration points. In an asymptomatically valid setup this implies that for $\tau_i(p)$ we have $\Pr[s(\boldsymbol{x}_i + \boldsymbol{\epsilon}, y_i) \geq \tau_i(p)] \geq p$. To account for finite samples we reduce $p$ to $p^\downarrow$ ($p^\downarrow \leq p$). Again this holds for each calibration with $1/(|\mathcal{D}_{\text{cal}}| + k)$ probability, and the conformal threshold is $(\tau_p, p^\downarrow)$. In the test time the setup is identical to the fixed-$\tau$ setup.

### D.3 REALISTIC SETUP TO EVALUATE GNN ROBUSTNESS

Concurrent to our work Gosch et al. (2023) shows that transductive setup is flawed to evaluate GNN robustness. This is since assuming that the clean graph is accessible to the defender during training, can lead to perfect robustness by simply remembering the clean graph. Many robust- and self-training GNNs proposed for robustness also exploit this flaw.

To evaluate GNN node-classification for robust CP, following prior works, we assumed the transductive setup where the clean graph is given during the training and calibration. Specifically, we assume that the defender trains, and computes the calibration scores on the clean graph. Perturbations in test nodes are then applied after calibration (in evasion setup). This again is based on similar assumption

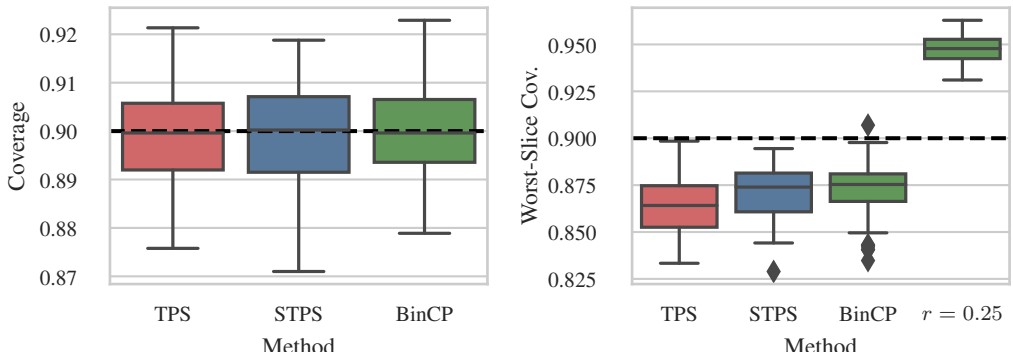

Figure 12: Comparison of coverage [Left] and worst-slice coverage [Right]. Here the STPS refers to the smooth TPS which is the average of 2000 randomly smooth inferences per point. The results are for CIFAR-10 dataset and r= 0 unless specified.

made by many other GNN robustness works. Therefore our robust node-classification results are only representing the comparison of BinCP and other approaches on a sparse binary certificate.

A more realistic setup is the inductive setup where the defender is given a clean subgraph $\mathcal{G}_{tr}$ for training (and calibration), and the adversary perturbs the rest of the graph upon arrival; i.e. the defender does not know the clean test graph. In vanilla setup, the conformal guarantee is still valid in the inductive setting via re-computing calibration nodes (Zargarbashi & Bojchevski, 2024). However in evasion, robust CP is not as easily applicable by recomputing the quantile and computing upper bounds on scores, since the calibration nodes are also affected by the adversarial test nodes through the message passing. Therefore robustness in GNNs under realistic setups is challenging which we leave it to future works.

# E ADDITIONAL EXPERIMENTS

**Various models and smoothing magnitudes ($\sigma$).** In Table 2 we compare the result between SOTA CAS and BinCP for CIFAR-10 dataset. The results are reported across various data smoothing $\sigma$ values, and models trained with different noise augmentations (data augmented during training with different $\sigma$ values). We call them smoothing $\sigma$, and model $\sigma$ respectively. In the robustness certificate for classification, it is considered best practice to use the same $\sigma$ in both model's noise augmentation, and the smoothing process. Similarly, in robust conformal prediction mismatching smoothing and model $\sigma$ results in a larger prediction set. Interestingly this adverse effect is much less observed in BinCP although it remains present. Overall, across all smoothing parameters, model $\sigma$ values, coverage rates, and perturbation radii, BinCP consistently outperforms CAS.

**Performance on small radii.** For completeness, in Fig. 9, we report the performance of BinCP on small values of $r$. As Jeary et al. (2024) reports $\sim 4.45$ average set size for $r = 0.02$ (Table 1 in (Jeary et al., 2024)) our report shows more than twice smaller sets for the same $r$. As in Table 2 we observe the same average set size for $r \sim 0.5$ ($\geq 20\times$ higher radius) for smallest $\sigma = 0.12$. As we discussed, one effect of this eye-catching difference is the inherent robustness of the randomized smooth prediction. As shown in Fig. 6, the empirical coverage of non-smooth prediction drastically decreases to 0 for small radii, while in smooth prediction the coverage decreases slowly.

**APS score function.** Although we observe similar comparison between CAS and BinCP given APS score function, for completeness we report the performance of both methods in Table 1.

**Conditional and class-conditional coverage.** Following Romano et al. (2020), we approximated the conditional coverage gap as the worst coverage among $n$ different slices. Each slice is defined as $\mathcal{X}_s = \{\boldsymbol{x}_i \in \mathcal{D} : a \leq \boldsymbol{x}_i \cdot \boldsymbol{v} \leq b\}$ for the random vector $\boldsymbol{v}$ and random scalers $a, b$ (Romano et al., 2020). For that, we sampled 200 random vectors $\boldsymbol{v}$ and among all the scalers randomly sampled $a, b$

Table 1: Comparison of smoothing-based robust CP methods on APS score

| | | $1 - \alpha = 0.9$ | | | | $1 - \alpha = 0.95$ | | | |
| | | CAS | | BinCP | | CAS | | BinCP | |
| $\sigma$ | r | Coverage | Set Size | Coverage | Set Size | Coverage | Set Size | Coverage | Set Size |
|---|---|---|---|---|---|---|---|---|---|
| | 0.06 | 0.954 | 1.635 | 0.946 | 1.529 | 0.990 | 4.022 | 0.980 | 2.151 |
| | 0.12 | 0.971 | 1.939 | 0.963 | 1.757 | 0.996 | 6.435 | 0.985 | 2.389 |
| | 0.18 | 0.987 | 2.879 | 0.978 | 2.076 | 1.000 | 9.745 | 0.991 | 2.876 |
| 0.12 | 0.25 | 0.998 | 7.454 | 0.986 | 2.510 | 1.000 | 10.000 | 0.995 | 3.405 |
| | 0.37 | 1.000 | 10.000 | 1.000 | 10.000 | 1.000 | 10.000 | 1.000 | 10.000 |
| | 0.50 | 1.000 | 10.000 | 1.000 | 10.000 | 1.000 | 10.000 | 1.000 | 10.000 |
| | 0.75 | 1.000 | 10.000 | 1.000 | 10.000 | 1.000 | 10.000 | 1.000 | 10.000 |
| | 0.06 | 0.955 | 2.108 | 0.944 | 1.894 | 0.986 | 3.316 | 0.976 | 2.677 |
| | 0.12 | 0.964 | 2.309 | 0.954 | 2.054 | 0.989 | 3.682 | 0.980 | 2.857 |
| | 0.18 | 0.970 | 2.495 | 0.961 | 2.227 | 0.993 | 5.038 | 0.986 | 3.181 |
| 0.25 | 0.25 | 0.980 | 2.900 | 0.972 | 2.537 | 0.997 | 6.004 | 0.989 | 3.444 |
| | 0.37 | 0.991 | 3.795 | 0.982 | 3.047 | 1.000 | 9.360 | 0.994 | 4.035 |
| | 0.50 | 0.999 | 7.430 | 0.991 | 3.729 | 1.000 | 10.000 | 0.997 | 4.850 |
| | 0.75 | 1.000 | 10.000 | 1.000 | 10.000 | 1.000 | 10.000 | 1.000 | 10.000 |
| | 0.06 | 0.956 | 2.738 | 0.942 | 2.479 | 0.981 | 3.864 | 0.975 | 3.342 |
| | 0.12 | 0.962 | 2.890 | 0.951 | 2.635 | 0.984 | 4.077 | 0.978 | 3.508 |
| | 0.18 | 0.968 | 3.078 | 0.959 | 2.801 | 0.986 | 4.277 | 0.981 | 3.658 |
| 0.50 | 0.25 | 0.973 | 3.304 | 0.966 | 2.994 | 0.989 | 4.546 | 0.984 | 3.899 |
| | 0.37 | 0.980 | 3.684 | 0.974 | 3.302 | 0.993 | 5.193 | 0.988 | 4.300 |
| | 0.50 | 0.986 | 4.153 | 0.979 | 3.663 | 0.996 | 5.868 | 0.991 | 4.733 |
| | 0.75 | 0.995 | 5.441 | 0.989 | 4.584 | 0.999 | 8.026 | 0.995 | 5.542 |

from the set $\{\boldsymbol{x}_i \cdot \boldsymbol{v}, \boldsymbol{x}_i \in \mathcal{D}\}$. We report the result over 100 different calibration samplings. In each iteration of the experiment, we exclude the slices with less than 200 points of support.

BinCP has smoothing, binarization and accounting for perturbation radius. For a better intuition we observe the effect of each step separately: vanilla TPS (without smoothing) as the baseline score, vanilla smooth TPS (labeled as STPS) which is an average of TPS scores over randomized sampled (same as CAS), BinCP without robustness (set $r = 0$) which reflects the effect of binarization (we did not correct for finite sample due to validity in vanilla setup), and robust CP via BinCP. As shown in Fig. 12, the smooth model has a better worst-slice coverage than vanilla TPS. Though binarization although the average worst-slice coverage remains the same, there is a slight decrease in the variance of this metric. Note that sampling correction and making CP robust increases the empirical coverage guarantee, therefore the worst slice coverage is increased due to the inherent increase in marginal coverage.

We also reported the result of the class-conditional coverage in Fig. 13. Empirically in almost all classes, BinCP is closer to the nominal guarantee compared to normal smoothing. Ultimately both smooth prediction and BinCP are not comparable with vanilla TPS.

**Comparison with RSCP+.** Yan et al. (2024) shows a flaw of RSCP (Gendler et al., 2021) indicating that the score function is not corrected for finite sample estimation. They show that by adding finite sample correction to RSCP, it becomes significantly inefficient and produces trivial sets $\mathcal{C}(\boldsymbol{x}_{n+1}) = \mathcal{Y}$. They remedy that by designing a ranking-based transformation on top of the given score function which defines a new score as

$$s_{\text{ppt}}(\boldsymbol{x}, y) = \sigma \left( \frac{1}{T|\mathcal{D}_{\text{tune}}|} \text{rank}(s(\boldsymbol{x}, y); \{s(\boldsymbol{x}_j, y_j)\}_{(\boldsymbol{x}_j, y_j) \in \mathcal{D}_{\text{tune}}}) - \frac{b}{T} \right) \quad (13)$$

Where $\mathcal{D}_{\text{tune}}$ is a holdout tuning index, $T$ is the temperature parameter, $b$ is a bias parameter, and $\sigma$ is the sigmoid function. The original experiment from Yan et al. (2024) has several issues, which we resolved and compared with it: (i) The scores have possible ties; i.e. two different data points can have the same score value. To remedy that we added an unnoticeable random number $\delta \sim \text{Uniform}[0, 1/|\mathcal{D}_{\text{tune}}|]$ to the scores. (ii) The tuning set and the calibration set in the experiments are significantly large. Yan et al. (2024) use 5250/10000 test datapoints as a tuning and calibration set. This unrealistic holdout labeled set contradicts the sparse labeling assumption. In our reproduction of their results we used a total of $\sim 380$ datapoints where 200 of them are for

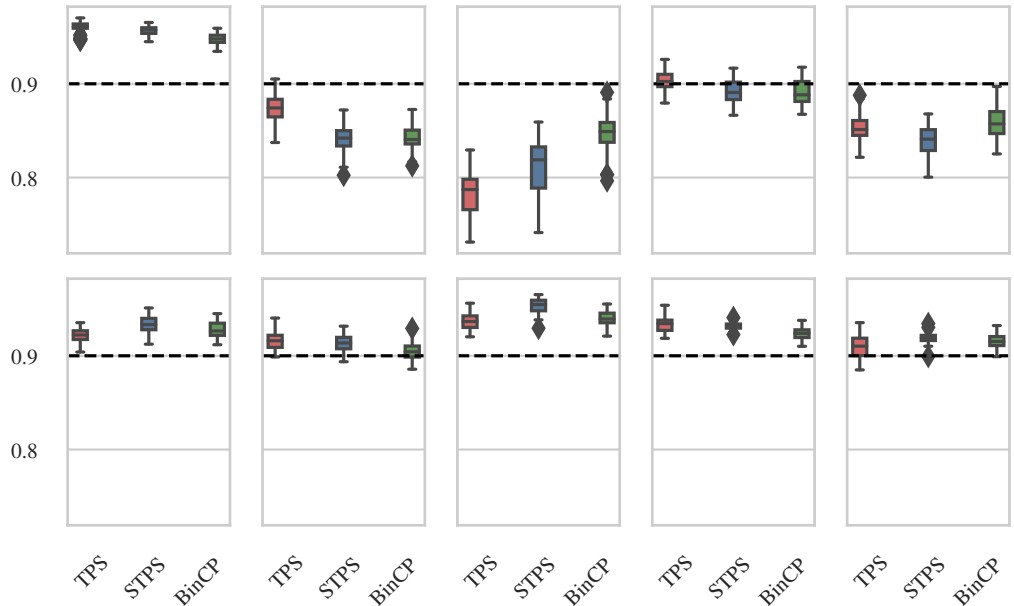

Figure 13: Comparison of methods in class-conditional coverage for all classes of CIFAR-10, note that here BinCP is used without sampling correction. That is because the correction slightly increases empirical coverage which can be misleading.

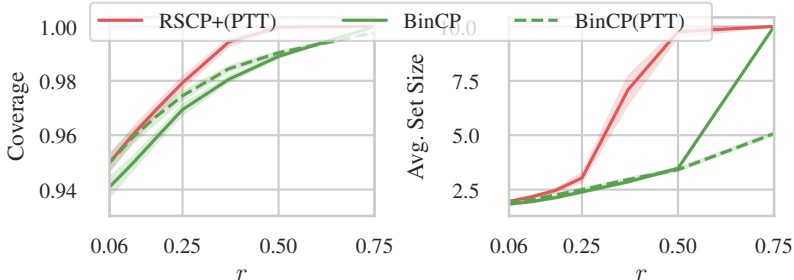

Figure 14: Comparison between BinCP and RSCP+ (PPT, Eq. 13) and BinCP with Eq. 13. The result is on CIFAR-10 dataset with $\sigma = 0.25$.

tuning. As shown in Fig. 14, BinCP still outperforms RSCP+(PPT). As the score function in Eq. 13 is also a valid score, we can use BinCP on top which shows slightly better efficiency for larger radii compared to BinCP combined with TPS score. Here we set $b = 1 - \alpha$, and $T = 0.001$, and report the results on 2000 MC samples. The reported result is on CIFAR-10 dataset.

| | | | $1-\alpha = 0.9$ | | | | $1-\alpha = 0.95$ | | | |
| | | | **CAS** | | **BinCP** | | **CAS** | | **BinCP** | |
| **Model** $\sigma$ | **Data** $\sigma$ | $r$ | Coverage | Set Size | Coverage | Set Size | Coverage | Set Size | Coverage | Ave Set Size |
|---|---|---|---|---|---|---|---|---|---|---|
| 0.12 | 0.12 | 0.06 | 0.950 | 1.581 | 0.942 | 1.483 | 0.987 | 3.353 | 0.976 | 2.009 |
| | | 0.12 | 0.968 | 1.839 | 0.959 | 1.671 | 0.996 | 6.731 | 0.986 | 2.387 |
| | | 0.18 | 0.985 | 2.761 | 0.974 | 1.946 | 0.999 | 9.417 | 0.990 | 2.666 |
| | | 0.25 | 0.997 | 7.078 | 0.985 | 2.369 | 1.000 | 10.000 | 0.994 | 3.213 |
| | 0.25 | 0.06 | 1.000 | 10.000 | 0.943 | 4.911 | 1.000 | 10.000 | 0.977 | 6.500 |
| | | 0.12 | 1.000 | 10.000 | 0.953 | 5.328 | 1.000 | 10.000 | 0.984 | 6.880 |
| | | 0.18 | 1.000 | 10.000 | 0.961 | 5.711 | 1.000 | 10.000 | 0.991 | 7.366 |
| | | 0.25 | 1.000 | 10.000 | 0.974 | 6.323 | 1.000 | 10.000 | 0.994 | 7.628 |
| | | 0.37 | 1.000 | 10.000 | 0.988 | 7.133 | 1.000 | 10.000 | 0.998 | 8.387 |
| | | 0.50 | 1.000 | 10.000 | 0.996 | 7.990 | 1.000 | 10.000 | 0.999 | 9.049 |
| | | 0.75 | 1.000 | 10.000 | 1.000 | 10.000 | 1.000 | 10.000 | 1.000 | 10.000 |
| | 0.50 | 0.06 | 1.000 | 10.000 | 0.950 | 8.836 | 1.000 | 10.000 | 0.980 | 9.310 |
| | | 0.12 | 1.000 | 10.000 | 0.960 | 8.985 | 1.000 | 10.000 | 0.984 | 9.402 |
| | | 0.18 | 1.000 | 10.000 | 0.965 | 9.075 | 1.000 | 10.000 | 0.986 | 9.450 |
| | | 0.25 | 1.000 | 10.000 | 0.974 | 9.222 | 1.000 | 10.000 | 0.990 | 9.535 |
| | | 0.37 | 1.000 | 10.000 | 0.983 | 9.403 | 1.000 | 10.000 | 0.996 | 9.656 |
| | | 0.50 | 1.000 | 10.000 | 0.991 | 9.557 | 1.000 | 10.000 | 0.999 | 9.789 |
| | | 0.75 | 1.000 | 10.000 | 0.999 | 9.820 | 1.000 | 10.000 | 1.000 | 9.947 |
| 0.25 | 0.12 | 0.06 | 0.954 | 2.570 | 0.937 | 2.232 | 0.991 | 7.016 | 0.968 | 2.992 |
| | | 0.12 | 0.969 | 3.049 | 0.949 | 2.395 | 0.998 | 9.042 | 0.976 | 3.301 |
| | | 0.18 | 0.984 | 4.573 | 0.956 | 2.562 | 1.000 | 9.845 | 0.981 | 3.567 |
| | | 0.25 | 0.999 | 9.510 | 0.969 | 2.908 | 1.000 | 10.000 | 0.986 | 3.895 |
| | 0.25 | 0.06 | 0.953 | 2.051 | 0.941 | 1.836 | 0.984 | 3.307 | 0.974 | 2.551 |
| | | 0.12 | 0.960 | 2.183 | 0.950 | 1.951 | 0.991 | 4.077 | 0.981 | 2.832 |
| | | 0.18 | 0.969 | 2.411 | 0.959 | 2.126 | 0.994 | 5.242 | 0.984 | 3.054 |
| | | 0.25 | 0.979 | 2.790 | 0.969 | 2.394 | 0.997 | 6.749 | 0.988 | 3.295 |
| | | 0.37 | 0.991 | 3.867 | 0.981 | 2.858 | 1.000 | 9.660 | 0.994 | 3.888 |
| | | 0.50 | 0.999 | 7.824 | 0.989 | 3.480 | 1.000 | 9.948 | 0.996 | 4.564 |
| | 0.50 | 0.06 | 1.000 | 10.000 | 0.947 | 6.762 | 1.000 | 10.000 | 0.979 | 7.837 |
| | | 0.12 | 1.000 | 10.000 | 0.956 | 7.024 | 1.000 | 10.000 | 0.983 | 8.016 |
| | | 0.18 | 1.000 | 10.000 | 0.960 | 7.212 | 1.000 | 10.000 | 0.988 | 8.221 |
| | | 0.25 | 1.000 | 10.000 | 0.969 | 7.523 | 1.000 | 10.000 | 0.992 | 8.430 |
| | | 0.37 | 1.000 | 10.000 | 0.981 | 7.935 | 1.000 | 10.000 | 0.996 | 8.763 |
| | | 0.50 | 1.000 | 10.000 | 0.990 | 8.350 | 1.000 | 10.000 | 0.998 | 9.040 |
| | | 0.75 | 1.000 | 10.000 | 0.997 | 9.008 | 1.000 | 10.000 | 0.999 | 9.494 |
| 0.50 | 0.12 | 0.06 | 0.948 | 3.701 | 0.923 | 3.060 | 0.994 | 8.485 | 0.965 | 4.196 |
| | | 0.12 | 0.961 | 4.230 | 0.929 | 3.159 | 0.999 | 9.564 | 0.971 | 4.457 |
| | | 0.18 | 0.980 | 5.843 | 0.937 | 3.330 | 1.000 | 9.960 | 0.973 | 4.538 |
| | | 0.25 | 0.998 | 9.329 | 0.947 | 3.550 | 1.000 | 10.000 | 0.977 | 4.753 |
| | 0.25 | 0.06 | 0.943 | 3.152 | 0.925 | 2.792 | 0.990 | 7.254 | 0.969 | 4.013 |
| | | 0.12 | 0.951 | 3.417 | 0.933 | 2.929 | 0.995 | 7.818 | 0.973 | 4.172 |
| | | 0.18 | 0.961 | 3.771 | 0.942 | 3.112 | 0.996 | 8.476 | 0.976 | 4.276 |
| | | 0.25 | 0.970 | 4.095 | 0.948 | 3.246 | 0.998 | 8.916 | 0.978 | 4.416 |
| | | 0.37 | 0.987 | 5.773 | 0.959 | 3.586 | 1.000 | 9.958 | 0.984 | 4.753 |
| | | 0.50 | 0.999 | 9.121 | 0.970 | 3.952 | 1.000 | 10.000 | 0.988 | 5.171 |
| | 0.50 | 0.06 | 0.957 | 2.767 | 0.943 | 2.428 | 0.984 | 3.995 | 0.974 | 3.288 |
| | | 0.12 | 0.964 | 2.948 | 0.949 | 2.558 | 0.986 | 4.165 | 0.977 | 3.405 |
| | | 0.18 | 0.968 | 3.071 | 0.955 | 2.673 | 0.988 | 4.351 | 0.980 | 3.542 |
| | | 0.25 | 0.974 | 3.272 | 0.962 | 2.835 | 0.991 | 4.850 | 0.983 | 3.806 |
| | | 0.37 | 0.982 | 3.721 | 0.973 | 3.182 | 0.993 | 5.160 | 0.986 | 4.044 |
| | | 0.50 | 0.987 | 4.215 | 0.980 | 3.524 | 0.997 | 6.439 | 0.990 | 4.511 |
| | | 0.75 | 0.996 | 5.708 | 0.987 | 4.285 | 1.000 | 9.287 | 0.994 | 5.316 |

Table 2: Comparison of CAS and BinCP for model trained with various smoothing $\sigma$, and input data with different smoothing $\sigma$. Results are for CIFAR-10 dataset.

