# OpenReview forum: "Robust Conformal Prediction with a Single Binary Certificate"
_ICLR.cc/2025/Conference — ICLR 2025 Poster_

### Official Review · Reviewer_qz9Z · 2024-11-01

**Soundness:** 3
**Presentation:** 3
**Contribution:** 3
**Rating:** 6
**Confidence:** 3

**Summary:**

The authors proposed a robust conformal prediction scheme using binarized smooth scores via thresholding. The binarized prediction sets are computationally efficient with fewer MC samples.

**Strengths:**

The binarized conformal prediction sets are easy to compute and takes simple form. The claim is supported with numerical experiments. Discussions and comparison with previous methods are sufficient.

Discussion: The authors addressed my concerns and I updated my score.

**Weaknesses:**

1. It is confusing in the abstract to use specific numbers (e.g. $10^4$ and $10^2$) without any given context. What is the dataset and experiment setup?
2. It is not clearly explained why nonbinarized robust CP require significantly more MC samples. Can the authors provide a theorem? Or if there is intuitive explanation?
3. The authors mentioned that working with binarized variables allows using the tighter concentration inequalities and Clopper & Pearson confidence intervals. However, this is not formally defined, and it's not clear why a good concentration bound will imply lower sample size.
4. The numerical experiments seem to suggest the method is indeed superior requiring lower MC sample size. However, the paper lacks discussion of why this occurs nor theoretical guarantees for sample size are given.

**Questions:**

See weakness.

---

> ### Author Response · Authors · 2024-11-19
>
> We thank the reviewer for their feedback. Here we address the weaknesses and questions:
>
> **W1.** We agree that without specifying the setup the numbers can be confusing, therefore we have rewritten the corresponding part denoting that the drop is reported on CIFAR-10 dataset. However, the higher efficiency with lower sample rate is consistent for all datasets.
>
> **W2 and W3.** In light of the reviewer’s comment we added Section E in the appendix that provides a high level introduction to randomized-smoothing based guarantees, and the effect of confidence intervals. Additionally, we supported the arguments with synthetic experiments which highlights the effect of sample size on each confidence interval. In short the answer is as following:
>
> As discussed in Section 5, robust CP in combination with probabilistic certificates requires us to compute either the true mean for RSCP, the true CDF for CAS, or the true Bernoulli parameter for BinCP. Since these values are unknown and computing them exactly is intractable we estimate them using Monte-Carlo (MC) sampling. For correctness of the certificate, we need to define confidence intervals that includes the true values with $1 - \eta$ probability (collectively) and account for this probability in calibration. The width of these confidence intervals (CIs) decreases with the number of MC samples — if a confidence interval is tighter, with fewer samples, we can attain the same (or smaller) width.
>
> Let $X \sim \mathcal{N}(0, 1)$ and $Y = f(X)$ for any arbitrary model f and $Z=Y>\tau$  for some constant $\tau$. To understand why binarization helps, think of $Y$ as the conformal score and $Z$ as the binarized score. Here $Z$ is a binary random variable with some unknown probability of success $p$. To estimate $p$ we can apply the *exact* Clopper Pearson CI. However, to estimate the mean of $Y$ we need to use the Hoeffding or Bernstein CI which provide wider (conservative) CIs. Since wider intervals increase the chance of a label to be in the prediction set, they directly effect the average set size.
>
> **W4.** In Section E we discuss the effect of sampling rate and the tightness of the confidence intervals on the average set size. A wider interval means a larger shift of the conformal quantile toward lower values. Similarly, at the test time confidence intervals shift the test score to higher values to ensure the validity of the certificate. Both of these shifts are increasing the chance of any label to be added to the prediction set, which means increase in the average set size. Additionally, the average of binary scores is often higher than the average of soft scores. To see this let $s_i$ be the soft score. Since  $b_i = s_i > \tau$ is bigger than $s_i$ (for a high enough $s_i$  or low enough  $\tau$), the average  $1/N \sum_i b_i$ is also bigger than  $1/N \sum_i s_i.$ Indeed, on our real-world data (CIFAR-10) the binarized average is bigger than the soft average in 89.8% cases (of true class for $\tau=0.3$). Bigger average implies better worst-case probability which implies smaller sets.

---

> > ### Comment · Reviewer_qz9Z · 2024-11-23
> >
> > Thanks for the authors' reply addressing my concerns and added sections for clarifications. I'm happy to update the score accordingly.
> >
> > I would still like to see theoretical guarantees for sample sizes. How much lower can the sample size be by using a binary certificate? It doesn't have to the most general for all distributions. As is mentioned in the paper, "e.g. For scores sampled from a Gaussian N (0.5, 0.1) Clopper Pearson error (for z ≥ 0.6) with 100 samples is still lower than Bernstein’s error with 250 samples." I would appreciate a proposition that exactly charaterizes the number of samples required in the Gaussian toy example.

---

> > > ### Author Response · Authors · 2024-11-27
> > >
> > > Thank you again for your feedback and for increasing our score. In the appendix (Section E), we added a new result (Proposition 3) that exactly quantifies when the Clopper-Pearson bound is better than the Hoeffding bound. We are happy to discuss this further if you have any questions.

---

### Official Review · Reviewer_mZko · 2024-11-03

**Soundness:** 2
**Presentation:** 2
**Contribution:** 4
**Rating:** 6
**Confidence:** 4

**Summary:**

The work presents a conformal prediction that further ensures adversarial robustness. This can be done by modifying the use of the conformity scores in either the calibration or prediction procedures, by the means of upper/lower bounds on particular adversarial objectives combined with inference of means (e.g. by Monte Carlo). This work further proposes that this adversarial objective should be of binary nature. By doing so they achieve tighter intervals and more efficient inference of the means (as the binary nature of the samples makes this estimation easier), and thus more efficient calibration/prediction.

**Strengths:**

The paper's contribution is solid. The proposed method has favorable theoretical properties, is widely applicable, and is empirically shown to significantly outperform previously available methods.

**Weaknesses:**

The paper is strong overall. Its weaknesses are, essentially:
(i) several typos, listed below;
(ii) some notation that needs to be made more consistent;
(iii) a couple of minor corrections necessary; and
(iv) a couple of confusing paragraphs.

Of these, of course, (iv) is the most severe.

(i) Typos:
- Line 088: 'finnaly'
- Line 096 and 144: the authors seem to use `\mapsto` for the 'function declaration' notation $f : X \to Y$; usually one uses just `\to`. (At least I don't think I've ever seen `\mapsto` used in this context).
- Line 161: missing a closing bracket on the indicator.
- Line 189: 'differ' -> 'defer'
- Line 223: missing an article before the $\bar{s}\_y(x) = [\cdots]$
- Lines 229-234: is the lack of the $y$ subscript intentional? I understand that what is being said in that paragraph is more general, but as far as I can tell it is then applied to $f_y$ (rather than some general $f$ or other function), and the lack of the subscript made this paragraph a bit confusing. Furthermore, the following paragraphs seem to act as if $f_y$ was used here...
- Line 236: $c^\uparrow$, not $c^\downarrow$
- Line 246: 'they are just need'

(ii) Notation:
- The paper currently mixes different notations for indicators (both $\mathbf{1}$ and $\mathbb{I}$). Pick one and stick with it.
- Line 238: $r$ was not properly introduced, as far as I can tell
- It's worth noting that the authors seem to use conformity scores 'flipped': in their use, a high conformity score would correspond to a high belief that $y$ conforms to $x$, whereas in the typical literature is the opposite. This doesn't need to be changed, but deserves to be explicitly said.

(iii) Minor corrections:
- The quantile calibration for Split Conformal Prediction requires a modified quantile, not just $\mathbb{Q}(\alpha; \{s\_i\}\_{i=1}^n)$. You need to either include $+\infty$ ($\mathbb{Q}(\alpha; \{s\_i\}\_{i=1}^n \cup {+\infty})$) or modify the level ($\mathbb{Q}(\alpha (1/n + 1); \{s\_i\}\_{i=1}^n)$ when the conformity score is the other way around, if I am not mistaken).
- Lines 129-131: perhaps could be a bit more formal? See the questions box.

(iv) Confusing paragraphs:
- I could not understand lines 246-250 (and thus the paragraphs that follow), and question their correctness -- or at least their presentation. Nevertheless, the work does seem not hinge on this particular point, so I don't consider this major (though it should be fixed/improved!).
- Lines 173-181 were also a bit confusing to read, but seemed reasonable. Either way, the theoretical result they are alluding to is formalized in Lemma 1, which is satisfyingly proven by Conformal Risk Control.

**Questions:**

- Lines 129-131: To be totally formal, this should use $\mathcal{B}^{-1}$, right?
- Line 238: what is $r$?
- Could the authors clarify lines 246-250?

*Score:* the paper carries a meaningful contribution and seems reasonably solid, and thus I lean towards acceptance. However, the presentation (and possibly a couple of paragraphs) need to be fixed/improved, which keep me from giving a higher score. Should they be fixed, I would be happy to increase my score accordingly.

---

> ### Author Response · Authors · 2024-11-19
>
> We thank the reviewer for reading our paper and the detailed discussion of the problems. We are happy that the reviewer liked our paper.
>
> **Readability.** In almost all mentioned cases we have updated the paper, however we wanted to highlight the following comments.
>
> **Lines 129-131.** Yes the reviewer is right. For the upper bound (test-time robustness) the ball is $\mathcal{B}^{-1}$. We corrected that. Thank you for mentioning. For the lower-bound (calibration-time robustness) the $\mathcal{B}$ ball is correct.
>
> **Lines 246-250.** The solution of Eq. 5 (the worst $\tilde{\boldsymbol{x}}$ and the worst $h$) is always at the border of the perturbation ball; e.g. for Gaussian smoothing $\\|\boldsymbol{x} - \tilde{\boldsymbol{x}}\\|_2 = r$. Additionally, the problem is rotation and translation invariant, meaning that we can move and rotate the coordinates to where we have $\boldsymbol{x} = [0, \dots, 0]$ and  $\tilde{\boldsymbol{x}} = [r, 0, \dots, 0]$. Therefore, the solution to Eq. 5, is not dependent to the point, and by design of the problem it is also not dependent to the definition of the function $f$. Therefore, we can write the lower bound just as a function of the ball, and the scalar $p$. In light of the comment we changed some parts of the text. We also refer to Section B where we discuss canonical points. This is discussed in detail both in [1] and [2].
>
> **Lines 173-181.** We provided this paragraph as another sketch of the proof aside the proof of Prop. 1 in the appendix. We agree that the discussion in these lines is confusing. We try to come up with some illustration for the camera ready version.
>
> **Line 238, the variable r.** This is the maximum possible magnitude of the perturbation, or simply the radius of the ball in the l2-norm threat model. We agree that it is better to define it explicitly. For that we modified line 113.
>
> **Line 229, function f.** Yes the reviewer is right. We intentionally did not write the subscript y. The definition in the following lines is for any general function f. However, we agree that this can be confusing. We added a description that this applies for all classes in the modified version.
>
> We again thank the reviewer for the time and comments on the paper which helps to make it stronger. In case that there is any missing point or further discussion we would be happy to receive follow-up comments.
>
> [1] Kumar, Aounon, et al. "Certifying confidence via randomized smoothing." *Advances in Neural Information Processing Systems* 33 (2020): 5165-5177.
>
> [2] Lee, Guang-He, et al. "Tight certificates of adversarial robustness for randomly smoothed classifiers." *Advances in Neural Information Processing Systems* 32 (2019).

---

> > ### Comment · Reviewer_mZko · 2024-11-25
> >
> > I'd like to thank the authors for their improvements, which have improved the paper substantially. I still have to re-check lines 246-250 -- which I will do soon, and confirm on OpenReview that everything is okay there -- but otherwise I'm content with the directions of the paper. Pending this final check, I will increase my score to an accept.

---

> > > ### Author Response · Authors · 2024-11-27
> > >
> > > Thank you again for your feedback. Regarding lines 246-250 we have added a reference to section E, and D (appendix) in line 255 - 256. There we discuss both the canonical view and a high-level introduction to randomized smoothing-based certificates. For a more detailed discussion, we recommend Lee et al (2019) as an excellent reference. In case there is any missing point we are happy to discuss further.
> > >
> > > Lee, G. H., Yuan, Y., Chang, S., & Jaakkola, T. (2019). Tight certificates of adversarial robustness for randomly smoothed classifiers. Advances in Neural Information Processing Systems, 32.

---

### Official Review · Reviewer_w6Hb · 2024-11-04

**Soundness:** 4
**Presentation:** 3
**Contribution:** 4
**Rating:** 8
**Confidence:** 4

**Summary:**

The paper introduces a new algorithm for robust conformal prediction. The algorithm boils down to a combination of monte-carlo sampling + binarization that yields a new conformal score; this score can then be inverted into a prediction set (also accounting for the fact that the test $\tilde{x}$ may be noisy). Experiments show this approach is an order of magnitude more sample efficient and substantially faster than the previous state-of-the-art.

**Strengths:**

* Empirical results are compelling in terms of runtime and sample size improvement.
* Well written.

**Weaknesses:**

* Paper readability could be improved for non-experts in TCS like myself, who are not acquainted with “certificates” and with the prior work of Zargarbashi et al.

The conditioning in (1) does not seem formally correct to me. (I’m aware there are some differences in language between TCS and statistics that may be at play here.) My understanding is that $\tilde x_{n+1} \in \mathcal{B}(x_{n+1})$ holds deterministically (i.e., with probability 1). So what is the point of the conditioning? Why would you ever even allow the possibility that $\tilde x_{n+1} \notin \mathcal{B}(x_{n+1})$ in your threat model?

Theorem 1 comes way out of the blue in terms of notation.

___

Zagarbashi et al. reference is broken (no year)

“First, we get robust coverage with only a single certificate computation, while at least one certificate per calibration (or test) point.” Typo. Also, I did not fully understand this.

“also satisfy Eq. 1 (calibration-time)” is this a typo in Theorem 1?

**Questions:**

“when we do need sample correction, working with binary variables allows us to use tighter concentration inequalities” This statement confuses me because this is rarely true. Binary random variables generally have the _worst_ concentration properties of all bounded random variables in [0,1], not the best.

---

> ### Author Response · Authors · 2024-11-19
>
> We thank the reviewer for reading our paper and for comments. We are happy that the reviewer liked our paper. Here we address the concerns:
>
> **On randomized smoothing.**  In light of the reviewer’s comment, we added a Section E (in the appendix) on a high level explanation of randomized smoothing-based certificate. The purpose of that section is just to deliver a high level intuition. In the end, for a detailed technical discussion we referred to highlighted works in randomized smoothing. We hope that the description can help to gain an intuition about the prior state of the art Zargarbashi et al. 2024. In case that the description needs to be improved we would be happy to hear from the reviewer again.
>
> **Definition of Robust CP.**  The reviewer is right. Since $\tilde{\boldsymbol{x}} _ {n+1} \in \mathcal{B}({\boldsymbol{x}} _ {n+1}) $ is deterministic the probability of this event is always 1. Therefore $\Pr[y _ {n+1} \in \bar{\mathcal{C}}(\tilde{\boldsymbol{x}} _ {n+1}) | \tilde{\boldsymbol{x}} _ {n+1} \in \mathcal{B} ({ \boldsymbol{x} } _ {n+1} ) ] $ equals
> $\Pr[ y _ {n+1} \in \bar{\mathcal{C}} (\tilde{\boldsymbol{x}} _ {n+1}) ,  \tilde{\boldsymbol{x}} _ {n+1} \in \mathcal{B}({\boldsymbol{x}} _ {n+1})] $.  In prior work there is the following equivalent definition  $\Pr[y _ {n+1} \in \bar{\mathcal{C}}(\tilde{\boldsymbol{x}} _ {n+1}) ,  \forall\tilde{\boldsymbol{x}} _ {n+1} \in \mathcal{B}({\boldsymbol{x}} _ {n+1})]$. This definition best captures the intuition that we want $1 - \alpha$ coverage for any perturbed point. We modified the notation based on the reviewer’s comment in the updated version.
>
> **Notations in Prop 1.** We agree that the Prop. 1 introduces many notations. However we tried to deliver an intuition about what each defined variable is encoding in the rest of Section 3. We also proposed an alternative sketch of the proof based on vanilla conformal prediction in lines 173 to 181 (“quantile view”).
>
> **Questions.** As discussed in Section 5, robust CP in combination with probabilistic certificates requires us to compute either the true mean for RSCP, the true CDF for CAS, or the true Bernoulli parameter for BinCP. Here by concentration inequality we specifically refer to confidence intervals (CIs) of the unknown quantitates that we need to estimate. For instance, let $x_i \sim \mathcal{N}(0, 1)$ for $i$ from $0$, to $n$. With $y_i = f(x_i)$, the $y_i$ values have a different continuous distribution. However, $z_i = \mathbb{I}[f(x_i) \ge \tau]$ is a Bernoulli distribution with unknown parameter $p$. Although we don’t know the distribution of the $y_i$’s, we can bound their mean with Hoeffding or Bernstein inequalities. The resulting confidence interval is conservative (wider). For the value $p$, we can estimate an **exact** confidence interval for $p$ with Clopper Pearson resulting in the smallest possible confidence interval.  In Section E, we have added two experiments that compare setups using different inequalities and study the effect of binarization as well. Our added experiments show that binarization and estimating the Bernoulli parameter helps, specially with lower sample rates.
>
> In case there is any further questions or points missed from the response we would be thankful if the reviewer points them out.

---

> > ### Comment · Reviewer_w6Hb · 2024-11-25
> >
> > Thanks! Looks great.
> > I'll maintain my score of accept.
> >
> > As a reference for future work, the authors may want to look at this paper: https://arxiv.org/abs/2310.07850
> >
> > The "random localization" technique there may be helpful for the line of work the authors are pursuing.

---

### Official Review · Reviewer_3Npc · 2024-11-04

**Soundness:** 3
**Presentation:** 3
**Contribution:** 3
**Rating:** 8
**Confidence:** 3

**Summary:**

This work attempts to solve the "robust conformal prediction" problem, where we seek to construct conformal sets that satisfy the desired coverage guarantee at test time under worse-case distribution shift  between calibration and test data (adversarial perturbation). The baseline in this direction (RSCP) uses test-time robustness, where in contrast, this paper builds on a previous result, Zargarbashi et al., which utilizes the CDF structure of the conformity scores and constructs CDF aware sets (CAS), outperforming RSCP. This work in particular, is able to improve on CAS by manipulating the distribution of "smoothed" conformity scores.
They are able to show improvements with respect to CAS in terms of 1. length-efficiency of the prediction sets, 2.  lower Monte Carlo sampling budget and faster computation of the prediction sets compared to CAS, 3. Removing the need for boundedness assumption on the conformity scores.

**Strengths:**

1. Attempts an important problem: uncertainty quantification with CP in the presence of adversarial perturbations.

2. This paper is able to address a major limitation of the SOTA methods ( CAS and RSCP ) by reducing the Monte-Carlo sampling budget.

3. The sets under RSCP/CAS are trivially large. This work is able to construct smaller prediction sets, and hence taking a step towards mitigating this sub-optimality of the current methods.

4. The experiments are extensive and support the author's claims.

5. The binarization of the distribution of the smooth scores is theoretically grounded.

**Weaknesses:**

1. This paper does not address conditional coverage guarantees, which is of high relevance.
2. limited in the scope of attacker motives. particularly this paper only addresses the problem when the attacker seeks to reduce empirical coverage, and does not consider alternative goals.

**Questions:**

1. Can the authors elaborate on the effect of binarizing the distribution of smooth conformity scores on class-conditional coverage. The results seem very attractive in that they reducing MC-sampling budget and give smaller prediction sets, and but there are no experiments showing the effects on class-conditional coverage. (Are there any inherent limitations posed when binarizing the distribution of smooth conformity scores? i.e any information lost?)

2. How does BinCP perform compared to RSCP+ ( Yan et al. 2024) which also propose a transformation of the smooth score and achieve smaller prediction set sizes than RSCP. In addition to line 467 and 51, could you address whether/how the calibration-time robustness (binCP) can be applied on top of/combined with test-time robustness (RSCP+), and if that is not the case, could you elaborate further on comparing the two methods in terms of how efficient each are in improving the size of the prediction sets, or the consistency of the improvement trends observed for each.

---

> ### Author Response · Authors · 2024-11-19
>
> #
>
> We thank the reviewer for reading our paper and the comments which helps to increase the quality of our work. Here we address the concerns.
>
> **Conditional coverage.** This is a very interesting point. Thank you. We can study clean conditional coverage, or adversarial conditional coverage. For all existing robust CP method, by increasing the radius r, we increase the clean marginal coverage (since the sets are conservative) which in turn increases clean conditional coverage. To better understand the effect of BinCP on the clean conditional coverage, we did an empirical study to answer several different questions. We included the empirical results in Section D (appendix, Fig. 10).
>
> To estimate clean conditional coverage we follow the procedure in Romano et al. 2020 that computes the worst-slice coverage gap. How to estimate conditional coverage under adversarial attacks is an open problem that has not been studied so far.
>
> First we ask how smoothing and binarization affects conditional coverage. In Fig. 10, we see that smoothing alone (without any robustness, i.e. $r=0$) decreases the coverage gap (compare TPS with STPS). Adding binarization on top, does not change the coverage gap. Next, we ask what is the effect of robustness (increasing the radius r)? As expected, with $r>0$,  the marginal coverage and the worst-slice conditional coverage increase.
>
> **Alternative attacker goals.** Other objectives, including changing the set size and decreasing conditional coverage are both important and interesting. We leave this for future work.
>
> **Class-conditional coverage.** We add empirical results on clean class-conditional coverage in Section D (appendix, Fig. 11). As in the previous question, we again consider the effect of smoothing and binarization. We see that for some classes there is better class-conditional coverage before smoothing (TPS), while for some classes it is better after smoothing (STPS). On the other hand, binarization (BinCP) always improves over STPS, i.e. it is closer to the nominal level compared the non-robust but smooth model (STPS).
>
> **Information loss.** Since binarization is not a 1-1 operation, strictly speaking there is information loss due to the data processing inequality. However, it is not clear whether the classical notion of (mutual) information is relevant in this case. Think of NNs that process raw data to learn some representations. Under the classical view, the representation always have less information, however, often they are more useful, e.g. fitting a linear classifier on the learned representation is better than fitting a linear classifier in the input space. This idea has been studied at depth in Yilun et al 2020, and Pimentel et al 2021 under the names of usable (predictive) information and Bayesian information.
>
> Similarly, in our case we should consider what information is useful for CP, e.g. information about the ranking of different data points. We leave this interesting question for future work and thank the reviewer for pointing it out.
>
> **Comparison with Yan et al, 2024 (RSCP+).**  Jeary et al, 2024 has shown (in Table 1 in their paper)  a strong improvement compared to RSCP+. We show a strong improvement to Jeary et al, 2024 in Section D. Fig. 9. Therefore, by transitivity, we expect BinCP to perform better than RSCP+. We are trying to run their method on the same setup and we will report the result in another comment.
>
> **Combining calibration-time and test-time robustness.** These two methods are two alternative variants that can not be combined. Specifically in Lemma 2 we show that calibration-time and test-time robustness are exactly equal after binarization. Similarly Zargarbashi et al. 2024 (CAS) show that the calibration and test time robustness are equivalent approaches (Fig 4 center in their original paper). Therefore a hybrid approach is equivalent to applying robustness guarantee on CP twice. Another intuition behind avoiding such combination is that BinCP only uses Clopper Pearson confidence intervals however Yan et al, 2024 need to add one additional Hoeffding bound which significantly effects the efficiency of the final sets. We have discussed the effect of different confidence intervals in section E.
>
> References:
>
> Yaniv Romano, Matteo Sesia, and Emmanuel J. Cand`es. Classification with valid and adaptive
> coverage. arXiv: Methodology, 2020.
>
> Linus Jeary, Tom Kuipers, Mehran Hosseini, and Nicola Paoletti. Verifiably robust conformal pre-
> diction, 2024.
>
> Xu, Yilun, et al. "A theory of usable information under computational constraints." *arXiv preprint arXiv:2002.10689*(2020).
>
> Pimentel, Tiago, and Ryan Cotterell. "A Bayesian framework for information-theoretic probing." *arXiv preprint arXiv:2109.03853* (2021).

---

> > ### Author Response · Authors · 2024-11-21
> >
> > Following the discussion, we have added our comparison with Yan et al, (2024) in the appendix (Fig. 12, Section D). Overall BinCP with baseline score (TPS) outperforms RSCP+ with PTT (their score function). In addition, since BinCP is a general framework capable of working with any (bounded or unbounded) score function we have also reported BinCP(PTT) in the same figure. Overall for smaller radii, we don't see any significant difference, however for larger radii their transformation seems to be helpful.

---

> > > ### Comment · Reviewer_3Npc · 2024-11-30
> > >
> > > I thank the authors for the additional experiments added in the appendix to address the conditional coverage study. I also thank the authors for including comparisons with RSCP+, I believe these experiments have further added clarity and highlighted the value of the work.
> > >
> > > I maintain my high score of 8.

---

### Meta-Review · Area_Chair_Ppyz · 2024-12-23

**Metareview:**

The core problem addressed in the paper is constructing robust conformal prediction sets that maintain coverage guarantees under adversarial input perturbations.
Traditional conformal prediction methods provide uncertainty quantification by producing prediction sets that include the true label with a specified probability. However, under adversarial attacks or worst-case distribution shifts, these methods often fail to maintain coverage or require computationally expensive techniques.

As a main contribution, it introduces BinCP, a method that converts conformity scores into a binary form by applying an adaptive threshold, effectively binarizing the scores while preserving the necessary statistical properties.
This binarization allows for the use of tighter confidence intervals, like the Clopper-Pearson bound, to estimate conformity, leading to smaller prediction sets and computational efficiency.

The approch is quite original and the reviewers appreciated the contributions.

I recommend an *accept*

**Additional Comments On Reviewer Discussion:**

The reviewers found the paper valuable for its innovative approach to robust conformal prediction, which improves efficiency and reduces the size of prediction sets while maintaining coverage guarantees. They appreciated the simplicity of the binarization technique, which requires fewer Monte Carlo samples and avoids assumptions about bounded scores, making it more practical and broadly applicable. Reviewers highlighted the strong experimental results that showed better performance compared to existing methods and noted the method's solid theoretical foundation. They raised questions about aspects like conditional coverage, the impact of binarization on information loss, and combining calibration-time and test-time robustness. The authors addressed these concerns thoroughly, adding experiments, clarifications, and theoretical insights in response. Overall, the reviewers agreed that the method’s simplicity, efficiency, and strong empirical and theoretical backing make it a valuable contribution to the field.

---

### Decision · Program_Chairs · 2025-01-22

Accept (Poster)